# Large-scale gene expression alterations introduced by structural variation drive morphotype diversification in *Brassica oleracea*

Xing Li [1,3], Yong Wang[1,3], Chengcheng Cai[1,2,3], Jialei Ji[1,3], Fengqing Han [1,3], Lei Zhang[1,3], Shumin Chen[1], Lingkui Zhang[1], Yinqing Yang [1], Qi Tang[1], Johan Bucher[2], Xuelin Wang[1], Limei Yang[1], Mu Zhuang[1], Kang Zhang [1] ✉, Honghao Lv [1] ✉, Guusje Bonnema [2] ✉, Yangyong Zhang [1] ✉ & Feng Cheng [1] ✉

*Brassica oleracea*, globally cultivated for its vegetable crops, consists of very diverse morphotypes, characterized by specialized enlarged organs as harvested products. This makes *B. oleracea* an ideal model for studying rapid evolution and domestication. We constructed a *B. oleracea* pan-genome from 27 high-quality genomes representing all morphotypes and their wild relatives. We identified structural variations (SVs) among these genomes and characterized these in 704 *B. oleracea* accessions using graph-based genome tools. We show that SVs exert bidirectional effects on the expression of numerous genes, either suppressing through DNA methylation or promoting probably by harboring transcription factor-binding elements. The following examples illustrate the role of SVs modulating gene expression: SVs promoting *BoPNY* and suppressing *BoCKX3* in cauliflower/broccoli, suppressing *BoKAN1* and *BoACS4* in cabbage and promoting *BoMYBtf* in ornamental kale. These results provide solid evidence for the role of SVs as dosage regulators of gene expression, driving *B. oleracea* domestication and diversification.

*Brassica oleracea*[1] vegetable crops are worldwide cultivated over a wide range of climate zones, occupying an annual planting area of 3.77 million hectares, yielding 96.39 million tons, with an estimated economic value of 16.12 billion USD (FAO, 2020, http://faostat.fao.org/). *B. oleracea* crops are rich in essential and diverse nutrients[2,3], including crucifer-specific glucosinolates with a wide range of biological activities and exhibit enormous diversity. For example, cabbage develops a leafy head; cauliflower and broccoli form enlarged arrested inflorescences (curds); brussels sprouts grow axillary heading buds along their stems; Chinese kale develops a succulent stem; kohlrabi develops a swollen tuberous stem. This morphotype diversity of *B. oleracea* provides a particular example of how different organs of a plant species can be the

[1]State Key Laboratory of Vegetable Biobreeding, Key Laboratory of Biology and Genetic Improvement of Horticultural Crops of the Ministry of Agriculture and Rural Affairs, Sino-Dutch Joint Laboratory of Horticultural Genomics, Institute of Vegetables and Flowers, Chinese Academy of Agricultural Sciences, Beijing, China. [2]Plant Breeding, Wageningen University and Research, Wageningen, The Netherlands. [3]These authors contributed equally: Xing Li, Yong Wang, Chengcheng Cai, Jialei Ji, Fengqing Han, Lei Zhang. ✉e-mail: zhangkang01@caas.cn; lvhonghao@caas.cn; guusje.bonnema@wur.nl; zhangyangyong@caas.cn; chengfeng@caas.cn

target of domestication, resulting in high-yielding crops with special edible products. Although several genomic selection signals associated with specific morphotypes were revealed by population resequencing studies[4,5], the genetic mechanism underlying this rapid evolution and domestication remains elusive.

*B. oleracea* has evolved from a mesohexaploidization event shared by all *Brassica* species tens of million years ago (MYA)[6,7]. Following that, the *Brassica* ancestor experienced extensive homoeologous gene fractionation (loss), characterized by subgenome dominance, with two recessive subgenomes (medium fractionated (MF1) and most fractionated (MF2)) losing more genes than that of the dominant subgenome (least fractionated (LF))[8]. Homoeologous genes are syntenic paralogs between the three subgenomes. The biased distribution of single-nucleotide polymorphisms (SNPs), small insertions/deletions (InDels) and structural variations (SVs) in the three subgenomes of both *B. oleracea* and *Brassica rapa* has been described[4,5,9]. *B. oleracea* was further characterized by its higher numbers of transposable elements (TEs) than *B. rapa*, resulting in a larger genome size[10]. TEs have been implicated in the occurrence of SVs[11,12].

Pan-genome studies combined with cataloging genetic variations are instrumental in genetically dissecting crop domestication, environmental adaptation and phenotype diversification[11,13–15]. Among genetic variations, SVs have emerged as important hidden variations that were largely unidentified and overlooked previously[4,16–18]. In tomato, SVs identified through a graph-based genome captured higher levels of heritability in a genome-wide association study (GWAS)[13], showing a better performance in resolving both the allelic and locus heterogeneity[12]. Moreover, SVs were found to be associated with gene expression changes, a factor influencing phenotypic variation in plants[11–13,15]. Genomic comparison across different species identified bulk conserved noncoding sequences (CNS) in promoter regions that were associated with transcriptional or post-transcriptional regulation of genes[19]. Recent studies also revealed the role of intra-specific variation of CNS on gene expression associated with variation in important traits[11,12,15].

Previously, a *B. oleracea* pan-genome including nine accessions was constructed using short-read sequencing technology, showing that nearly 20% of genes are affected by presence/absence variation (PAV)[20]. Meanwhile, whole-genome comparison between five *B. oleracea* high-quality genomes assembled by long-reads further revealed extensive small-scale SNPs, InDels and large-scale SVs[5,21–23]. To capture the full genetic variation within *B. oleracea* population and investigate the genomic factors underlying its domestication and evolution, the construction of a high-quality pan-genome with a more comprehensive representation of morphotypes is strongly needed.

In this study, we de novo assembled chromosome-level genomes of 22 representative *B. oleracea* accessions, constructed a pan-genome and a graph-based genome using these 22 plus five previously reported high-quality genomes and determined genomic variations in a *B. oleracea* population of 704 accessions. By analyzing leaf mRNA-seq data of a core collection of 223 *B. oleracea* accessions, we revealed that SVs introduced large-scale gene expression alterations, affecting gene expression bidirectionally. Interestingly, SV-mediated gene expression alterations were under selection and significantly associated with specific morphotypes. These findings underscore the important role of SVs as expression dosage regulators of target genes. These bulk transcriptional variations introduced by SVs likely resulted in phenotypic diversity that is subject to domestication selection, leading to the success of the very diverse *B. oleracea* species.

## Results

### High-quality genome assembly of representative morphotypes

To construct a pan-genome that encompasses the full range of genetic diversity in *B. oleracea*, we analyzed the resequencing data of 704 globally distributed *B. oleracea* accessions covering all different morphotypes and their wild relatives (Supplementary Tables 1 and 2). We identified 3,792,290 SNPs and 528,850 InDels in these accessions using cabbage JZS as reference genome[22]. A phylogenetic tree was then constructed using SNPs, which classified the 704 accessions into the following three main groups: wild *B. oleracea* and kales, arrested inflorescence lineage (AIL) and leafy head lineage (LHL; Fig. 1a and Supplementary Note 2). The phylogenetic relationship revealed in our study was generally consistent with those reported previously[4,5,24,25]. Based on the phylogeny and morphotype diversity, we selected 22 representative accessions for de novo genome assembly (Table 1).

We assembled genome sequences of the 22 accessions by integrating long-reads (PacBio or Nanopore sequencing), optical mapping molecules (BioNano) or high-throughput chromosome conformation capture data (Hi-C) and Illumina short-reads (Methods; Supplementary Note 2 and Supplementary Tables 3–7). The total genome size of these assemblies ranged from 539.87 to 584.16 Mb with an average contig N50 of 19.18 Mb (Table 1). An average of 98% contig sequences were anchored to the nine pseudochromosomes of *B. oleracea*. The completeness of these genome assemblies was assessed using benchmarking universal single-copy orthologs (BUSCO), with an average of 98.70% complete score in these genomes (Supplementary Table 8).

To minimize artifacts that could arise from different gene prediction approaches, we predicted gene models of both the 22 newly assembled genomes and the five reported high-quality genomes[5,21–23] using the same annotation pipeline (Methods). Using an integrated strategy combining ab initio, homology-based and transcriptome-assisted prediction, we obtained a range of 50,346 to 55,003 protein-coding genes with a mean BUSCO value of 97.9% in these genomes (Table 1). After gene prediction, a phylogenetic tree constructed based on single-copy orthologous genes clustered the 27 genomes into three groups, similar to the results observed in the population (Fig. 1a and b).

A total range of 53.5–58.5% sequences in these *B. oleracea* genomes were TEs, with long terminal repeat retrotransposons (LTR-RTs) being the most abundant type (Supplementary Note 2). We further identified 4,703 to 6,253 full-length LTR-RTs (fl-LTRs) in these genomes (Supplementary Table 9), with recently inserted fl-LTRs enriched in centromeric regions (Fig. 1c). We revealed continuous expansion of Copia and

**Fig. 1 | Phylogenetic analysis and transposable element characteristics in *B. oleracea*. a**, Phylogenetic tree of 704 *B. oleracea* accessions. Different colors of branches indicate accessions from different morphotype groups. The images of the 27 representative accessions were placed next to their branches. The light blue, yellow and green backgrounds denote the following three main clusters: the wild/ancestral group, the arrested inflorescence lineage and the leafy head lineage. The red stars denote the 22 newly assembled genomes and the red rectangles denote five previously reported genomes. **b**, Phylogenetic tree of the 27 representative *B. oleracea* accessions, with the genome of *B. rapa* as the outgroup. **c**, The estimated insertion time (*y* axis) of all the full-length LTRs in the 27 *B. oleracea* genomes along the nine chromosomes (*x* axis) of *B. oleracea*. The lengths of nine chromosomes were normalized to 0–100, proportional to their physical lengths. Each dot represents one LTR insertion event. The heatmap denotes the density of the full-length LTRs. Purple bars below each chromosome denote centromeric regions detected by centromere-specific repetitive sequences. **d**, Distribution of insertion time of full-length Copia and Gypsy LTRs in the 27 individual genomes. Each line represents a genome in the left graph. The two circles show the Copia and Gypsy LTRs that can be clustered into groups with sequence similarity of ≥90%. **e**, The heatmap shows the TAD prediction on chromosome eight of T10 (as an example), in which the region colored in dark red denotes a TAD structure. The line charts below the heatmap show the density of Copia and Gypsy LTRs, respectively, highlighting the enrichment of Copia LTRs in the centromere region, which is surrounded by high density of Gypsy LTRs.

Gypsy in all the genomes since four MYA (Fig. 1d). In addition, Copia TEs were clustered into more and larger groups than Gypsy based on sequence similarity (Fig. 1d), suggesting that Copia was under stronger expansion than Gypsy. More than 80% of the centromeric sequences were annotated as Copia in *B. oleracea* (Fig. 1e and Supplementary Fig. 1). Interestingly, these enriched Copia islands in centromeres were

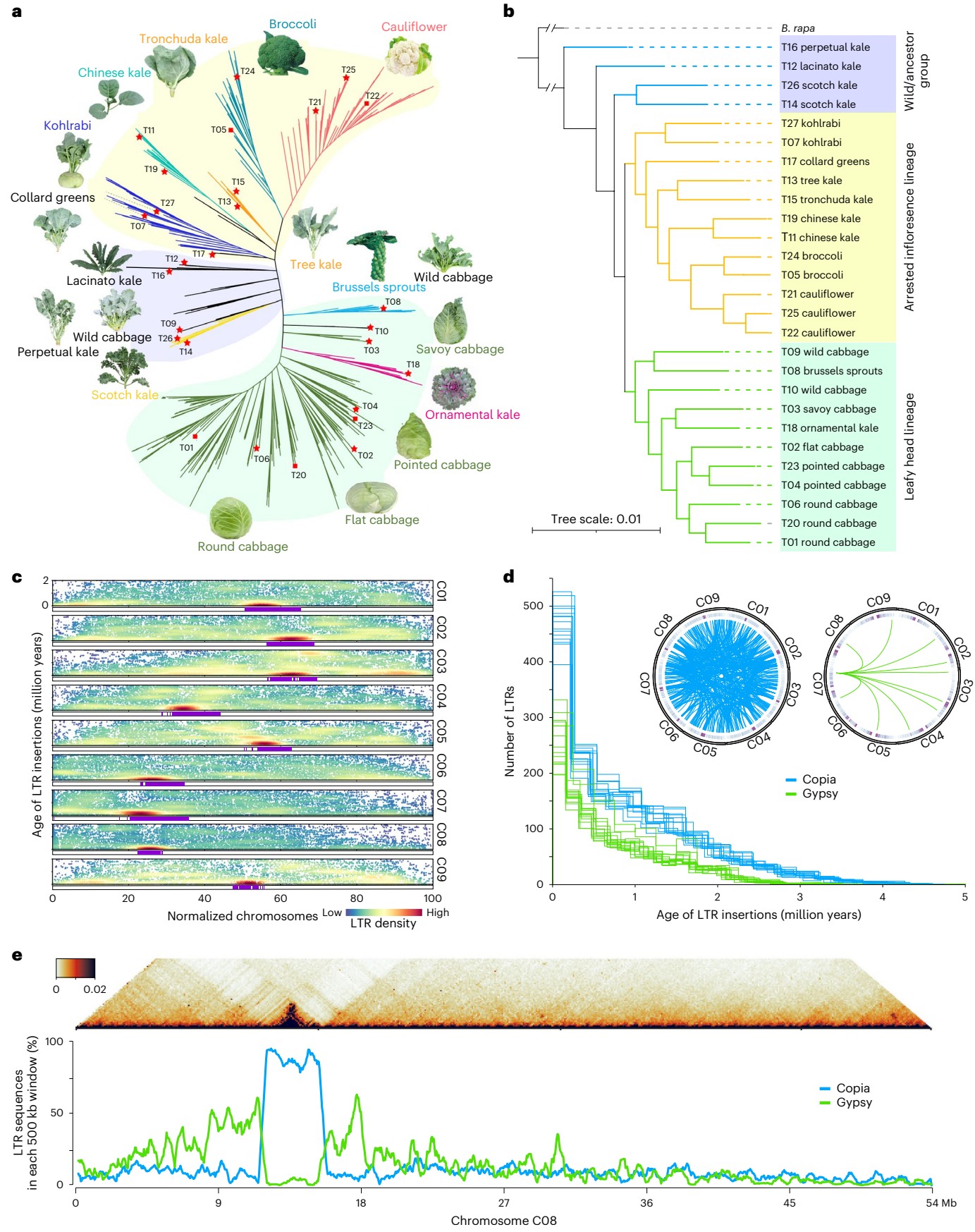

**Table 1 | Assembly and annotation metrics of the 27 *B. oleracea* genomes**

| Index | ID | Type | Latin name | Contig N50 (Mb) | Number of contig | Assembly size (Mb) | Anchored (%) | Number of genes | BUSCO (%) | TEs (%) |
|---|---|---|---|---|---|---|---|---|---|---|
| 1 | T02 | Cabbage | *B. oleracea* var. *capitata* | 17.13 | 159 | 558.94 | 99.49 | 52,459 | 98.60 | 60.32 |
| 2 | T03 | Cabbage | *B. oleracea* var. *capitata* | 31.30 | 152 | 570.74 | 99.15 | 52,881 | 97.10 | 60.95 |
| 3 | T04 | Cabbage | *B. oleracea* var. *capitata* | 33.13 | 99 | 578.71 | 99.06 | 53,336 | 98.20 | 60.85 |
| 4 | T06 | Cabbage | *B. oleracea* var. *capitata* | 12.17 | 316 | 565.47 | 94.36 | 51,050 | 98.70 | 56.82 |
| 5 | T07 | Kohlrabi | *B. oleracea* var. *gongylodes* | 7.12 | 363 | 549.02 | 99.76 | 52,968 | 98.30 | 58.41 |
| 6 | T08 | Brussels sprouts | *B. oleracea* var. *gemmifera* | 30.57 | 92 | 573.42 | 98.61 | 52,708 | 98.20 | 60.65 |
| 7 | T09 | Wild cabbage | Wild *B. oleracea* | 25.08 | 113 | 576.02 | 99.30 | 52,059 | 98.20 | 60.75 |
| 8 | T10 | Wild cabbage | Wild *B. oleracea* | 31.59 | 90 | 580.97 | 98.19 | 52,607 | 98.60 | 61.36 |
| 9 | T11 | Chinese kale | *B. oleracea* var. *alboglabra* | 24.91 | 108 | 565.57 | 99.03 | 51,827 | 97.50 | 60.44 |
| 10 | T12 | Lacinato Kale | *B. oleracea* var. *palmifolia* | 24.72 | 118 | 551.97 | 99.91 | 53,411 | 97.90 | 59.53 |
| 11 | T13 | Tree cabbage | *B. oleracea* var. *longata* | 10.01 | 245 | 552.22 | 99.85 | 52,326 | 98.40 | 58.98 |
| 12 | T14 | Curly kale | *B. oleracea* var. *sabellica* | 12.34 | 206 | 542.38 | 99.92 | 52,719 | 97.90 | 58.88 |
| 13 | T15 | Tronchuda kale | *B. oleracea* var. *costata* | 19.76 | 149 | 539.87 | 99.74 | 52,561 | 96.70 | 58.49 |
| 14 | T16 | Perpetual kale | *B. oleracea* var. *ramosa* | 12.02 | 376 | 562.15 | 99.89 | 53,406 | 97.70 | 59.59 |
| 15 | T17 | Collard greens | *B. oleracea* var. *viridis* | 2.18 | 1,083 | 584.16 | 97.95 | 55,003 | 94.40 | 58.80 |
| 16 | T18 | Ornamental kale | *B. oleracea* var. *acephala* | 26.13 | 136 | 553.36 | 99.47 | 53,089 | 98.50 | 59.80 |
| 17 | T19 | Chinese kale | *B. oleracea* var. *alboglabra* | 28.81 | 119 | 572.04 | 99.20 | 52,663 | 98.80 | 60.87 |
| 18 | T21 | Cauliflower | *B. oleracea* var. *botrytis* | 7.69 | 205 | 534.41 | 98.54 | 51,133 | 98.30 | 56.65 |
| 19 | T24 | Broccoli | *B. oleracea* var. *italica* | 14.46 | 316 | 558.04 | 94.26 | 50,938 | 98.70 | 59.12 |
| 20 | T25 | Cauliflower | *B. oleracea* var. *botrytis* | 11.43 | 270 | 547.89 | 95.77 | 50,759 | 98.00 | 55.50 |
| 21 | T26 | Curly kale | *B. oleracea* var. *sabellica* | 16.28 | 364 | 568.26 | 94.19 | 51,552 | 98.30 | 56.79 |
| 22 | T27 | Kohlrabi | *B. oleracea* var. *gongylodes* | 11.74 | 283 | 557.26 | 95.43 | 50,346 | 98.20 | 57.01 |
| 23 | T01[a] | Cabbage | *B. oleracea* var. *capitata* | 3.59 | 902 | 574.91 | 92.17 | 52,909 | 98.90 | 55.92 |
| 24 | T05[a] | Broccoli | *B. oleracea* var. *italica* | 9.49 | 264 | 554.98 | 95.29 | 51,934 | 98.60 | 53.46 |
| 25 | T20[a] | Cabbage | *B. oleracea* var. *capitata* | 2.37 | 1,184 | 561.15 | 96.17 | 53,113 | 97.10 | 57.81 |
| 26 | T22[a] | Cauliflower | *B. oleracea* var. *botrytis* | 4.97 | 615 | 552.84 | 99.04 | 51,028 | 95.60 | 57.06 |
| 27 | T23[a] | Cabbage | *B. oleracea* var. *capitata* | 3.10 | 973 | 565.47 | 95.34 | 52,649 | 98.10 | 58.43 |

[a]Previously reported genomes.

surrounded by high densities (>50%) of Gypsy in all the nine chromosomes of *B. oleracea*. Moreover, the topologically associating domain (TAD) structures overlapped with the Copia islands in all nine centromeric regions (Supplementary Fig. 1). This pattern was also found in six of ten chromosomes in *B. rapa* (Supplementary Fig. 2). These results suggest that Copia has an important role in the organization or function of centromeres through maintaining TAD structures.

**Homoeologous gene retention variation**

We constructed an orthologous pan-genome comprising the 27 *B. oleracea* genomes. In total, we identified 57,137 orthologous gene families using OrthoFinder[26] (Supplementary Note 3 and Supplementary Fig. 3). To investigate the retention variation of homoeologous genes among these mesohexaploid *B. oleracea* genomes, we further performed syntenic orthologous gene analysis (hereafter referred to as 'syntenic pan-genome'). In the orthologous pan-genome, homoeologs were assigned to one orthologous family, whereas syntenic pan-genome separates them into different syntenic gene families. We detected a total of 87,444 syntenic gene families based on genomic synteny among these genomes of which 32,721, 24,902 and 22,423 families were located at LF, MF1 and MF2 subgenomes, respectively. The number of syntenic gene families increased when adding additional genomes and approached a plateau when *n* = 25 (Fig. 2a), consistent with that of the orthologous

pan-genome. We further separated all these syntenic gene families into 20,306 (23.2%), 10,086 (11.5%), 55,205 (63.1%) and 1,847 (2.1%) syntenic core, softcore, dispensable and private gene families, respectively, with a mean of 21,680 (41.5%), 10,724 (20.5%), 17,236 (32.9%) and 2,675 (5.1%) per genome (Fig. 2b–d). We found significantly more TE insertions in syntenic dispensable and private genes than in syntenic core and softcore genes (Fig. 2e), suggesting that TEs contribute to genetic variations in these genes. The expression levels of syntenic core and softcore genes were significantly higher than those of syntenic dispensable and private genes (Fig. 2f). Moreover, the $K_a/K_s$ values of the syntenic core genes were significantly lower than that of the orthologous core genes (Supplementary Fig. 4b), supporting more conservation of the syntenic core genes. We found that 44.6% of syntenic private and 38.2% of syntenic dispensable genes belong to orthologous core and softcore genes (Supplementary Fig. 4a), respectively. This illustrates the extensive differential gene loss of homoeologs during the evolution and diversification of *B. oleracea*.

We dived into genes that were prone to being lost in different lineages/morphotypes of *B. oleracea*. A total of 20,924 syntenic gene families were lost in one to 14 genomes, while they were retained in 15 to 27 genomes. Among these, 2,786 and 5,139 gene families were lost exclusively in LHL and AIL, respectively (Fig. 2g). Intriguingly, in AIL, 556 syntenic gene families with gene loss specifically in broccoli

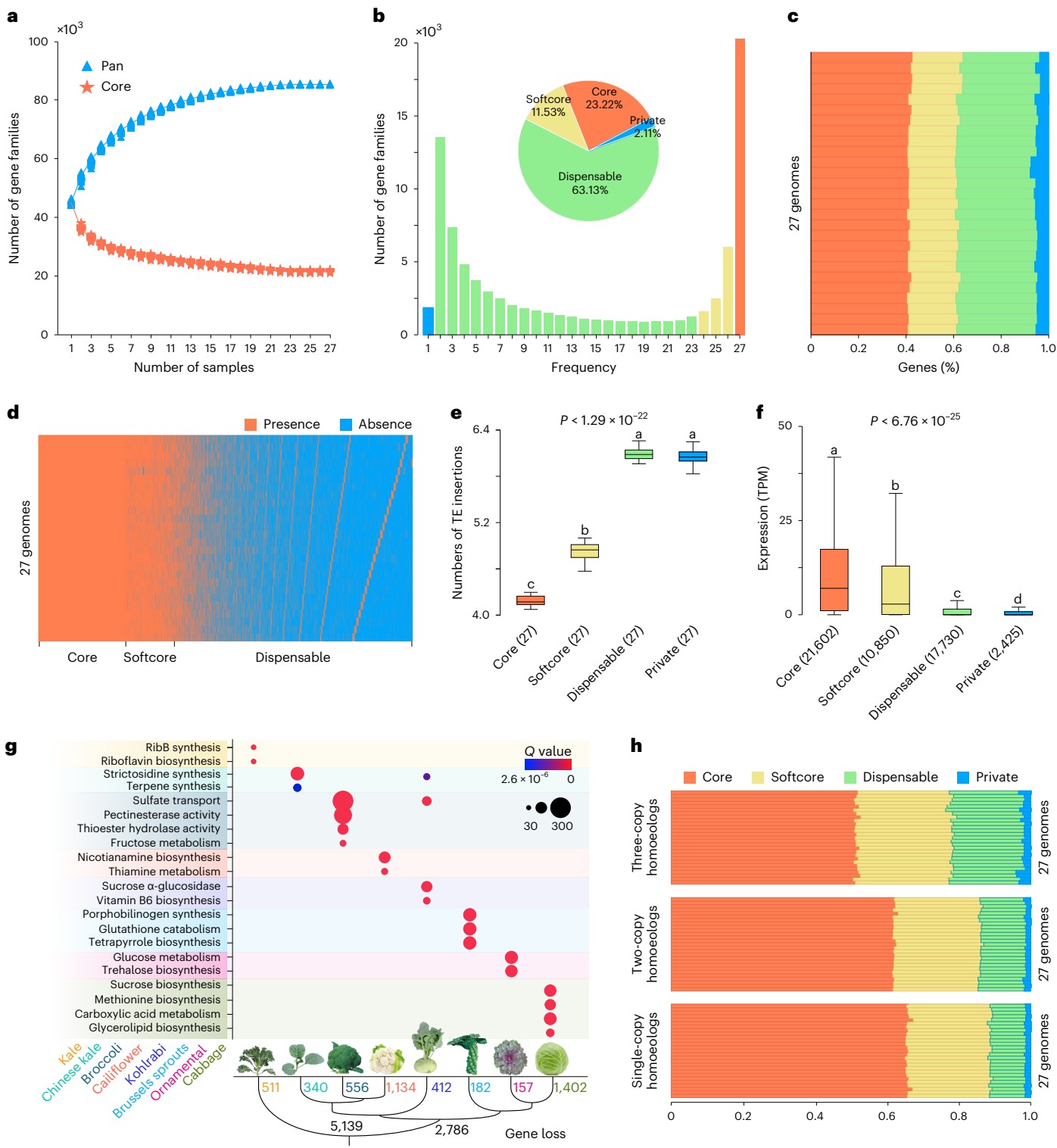

**Fig. 2 | The syntenic pan-genome constructed from the 27 *B. oleracea* genomes. a**, The number of syntenic pan and core gene families in the 27 genomes. **b**, Composition of the syntenic pan-genome. The histogram shows the frequency distribution of syntenic gene families shared by different numbers of genomes. The pie chart shows the proportion of different groups of syntenic gene families. **c**, Percentage of different groups of syntenic gene families in each of the 27 genomes. **d**, Presence and absence information of all syntenic gene families in the 27 genomes. **e**,**f**, The average number of TE insertions in genes and the expression level of genes in different groups of syntenic gene families (two-sided Student's *t* test; centerline, median; box limits, first and third quartiles; whiskers, 1.5× IQR). Different lowercase letters above the box plots represent significant differences (*P* < 0.05). **g**, Functional analysis (gene ontology) of lost genes in the syntenic softcore or dispensable gene families, in different *B. oleracea* morphotypes, highlighting strong function enrichment associated with specific metabolites. The number of lost genes in different morphotypes is provided in the tree diagram. **h**, Syntenic gene families were separated into three groups corresponding to the numbers of homoeologs (single-, two- or three-copy) retained from the *Brassica* mesohexaploidization event. The percentage of gene families in different pan-genome classes for these groups is shown in each of the 27 *B. oleracea* genomes. IQR, interquartile range.

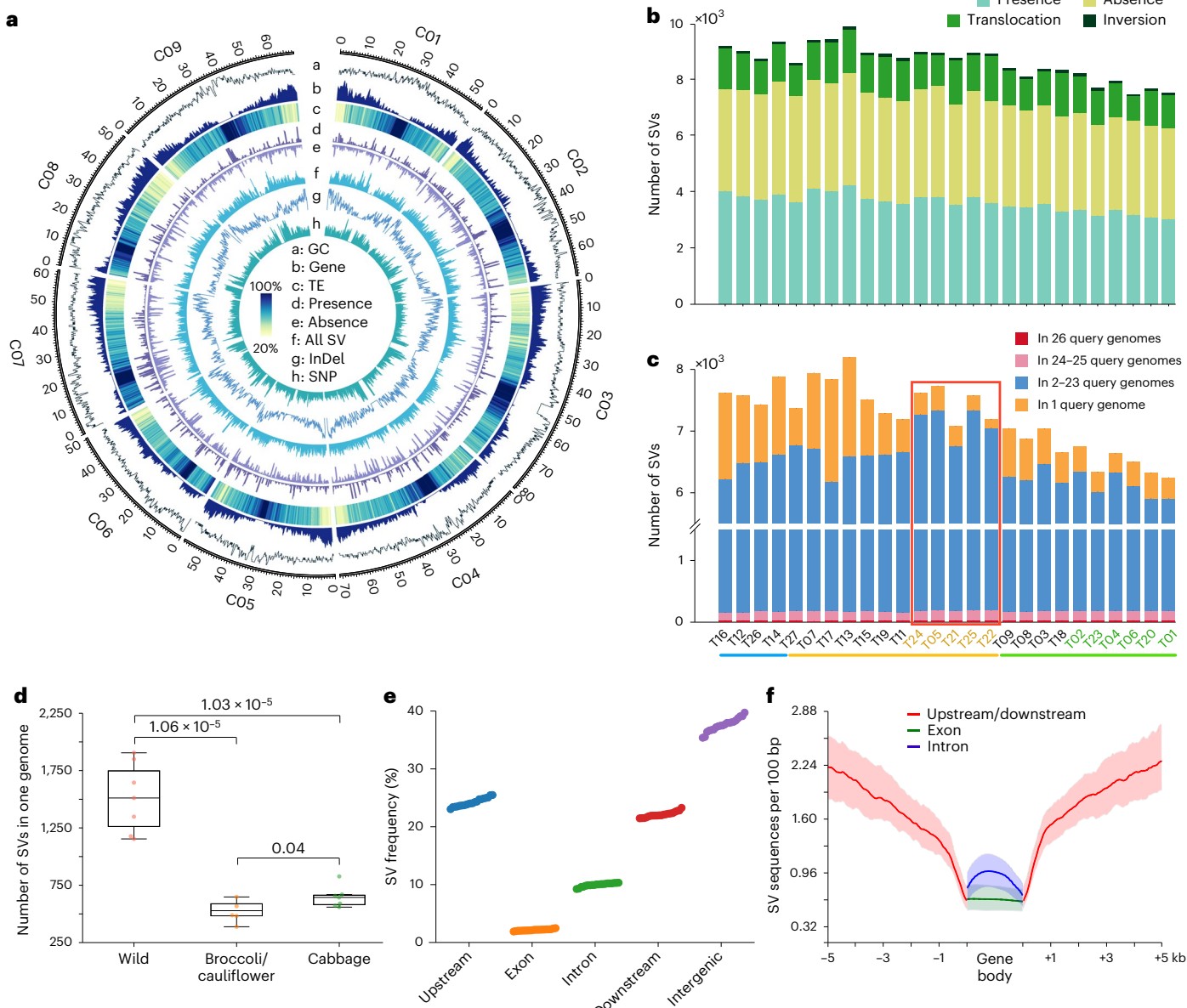

**Fig. 3 | Characteristics of structural variations identified in the 27 _B. oleracea_ genomes. a**, The distribution of GC content (33–41%), gene numbers (0–200 Mb⁻¹) and TE density (20–100%) in the T10 reference genome, the nonredundant SVs (presence, 2–100 kb/Mb; absence, 20–400 kb/Mb and all SVs, 10–400 kb Mb⁻¹) among 27 genomes, as well as the SNPs (10–40 kb⁻¹) and InDels (10–30 kb Mb⁻¹) identified in the 704 _B. oleracea_ accessions. **b**, The number of different types of SVs from the nonredundant set of SVs in individual _B. oleracea_ genomes. **c**, The number of SVs present in different numbers of query genomes. The bottom lines colored in light blue, light orange and light green mark these accessions from the wild/ancestral group, the AIL and the LHL, respectively. The sample IDs colored in light orange and light green denote accessions from broccoli/cauliflower and cabbage, respectively. The red rectangle marks the accessions of broccoli/cauliflower, highlighting the lower number of SVs in

broccoli/cauliflower compared to the other accessions. **d**, The number of private SVs in wild _B. oleracea_, broccoli/cauliflower and cabbage genomes, showing significantly more private SVs in wild _B. oleracea_ than in others ($n = 7$ versus 5 versus 7; two-sided Wilcoxon rank-sum test; centerline, median; box limits, first and third quartiles; whiskers, 1.5× IQR). **e**, The frequency distribution of SVs in the following five different genomic regions: upstream (within −3 kb), exon, intron, downstream (within +3 kb) and intergenic regions. The SV ratios in the five regions were calculated for each of the 27 genomes, and these values were then sorted and plotted from small to large for each of the five regions. **f**, The density of SV sequences per 100 bp in gene bodies and 5 kb flanking regions in the 27 _B. oleracea_ genomes. The area plots mark the maximum and minimum values across the 27 _B. oleracea_ genomes, and the lines denote average values.

were enriched in functions of sulfate transport, thioester hydrolase activity and riboflavin biosynthesis. In comparison, 1,134 syntenic gene families with gene loss specifically in cauliflower were enriched in nicotinamide biosynthesis and thiamine metabolism. Similarly, syntenic gene families with gene loss only in specific LHL morphotypes were found to be enriched in functions related to specific metabolites (Fig. 2g). The observations that genes specifically lost in different morphotypes were enriched in functions of biosynthesis or metabolism

of various nutrient contents, pointing to unique nutritional composition or flavor of specific _B. oleracea_ crops. In addition, our analysis of homoeologous copy-number variation (CNV) among _B. oleracea_ morphotypes revealed morphotype-specific loss of homoeologous genes, which may contribute to the evolution of these morphotypes through variation in gene copy dosage that is associated with expression dosage (Fig. 2h, Supplementary Note 3 and Supplementary Tables 10 and 11).

## Structural variations between the 27 *B. oleracea* genomes

The 27 high-quality *B. oleracea* genomes provide essential resources for the accurate identification of large-scale SVs. We aligned the sequences of 26 *B. oleracea* genomes to the T10 reference genome using Nucmer[27]. A total of 502,701 SVs were identified using SyRI[28], including 452,148 PAVs (50 bp to 3.34 Mb), 13,090 CNVs (50 bp to 243.14 kb), 2,263 inversions (1,022 bp to 12.18 Mb) and 35,200 translocations (9,002 intrachromosomal and 26,198 interchromosomal translocations; 505 bp to 5.59 Mb; Fig. 3a and Supplementary Fig. 5a). We randomly selected 30 large SVs (>8 kb) and 30 short SVs (<8 kb) for validation. Approximately 93% of the selected large SVs were validated by Hi-C paired-end reads; the remaining 7% could not be validated (Supplementary Fig. 6). For the selected short SVs, 97% were validated by long-reads; the remaining 3% were found to be false calls (Supplementary Fig. 7 and Supplementary Table 12).

We merged the 502,701 SVs into 56,697 nonredundant SVs. The number of these SVs ranged from 7,449 to 9,848 per genome (Fig. 3b). A total of 50,153 nonredundant PAVs were used in our subsequent analysis. Similar to that of orthologous and syntenic gene families, the number of SVs increased when adding additional genomes; this increase diminished when *n* = 25 (Supplementary Fig. 5b). Modeling this increase[29] predicts a total SV number of 58,410 ± 1,452. The number of shared SVs sharply declined for the first three genomes and slowly decreased thereafter. We identified 27 SVs present in all 26 query genomes, 168 SVs present in 24–25 query genomes, 26,641 SVs present in 2–23 query genomes and 18,226 SVs present in only one query genome, opposite to the trend of gene family counts (Fig. 3c). The number of private SVs in wild *B. oleracea* is significantly higher than in broccoli/cauliflower and cabbage, indicating extensive loss of genetic diversity during domestication of *B. oleracea* (Fig. 3c and d).

## SVs introduce expression variation in numerous genes

SVs distributed preferably in upstream and downstream regions of genes compared to gene bodies (Fig. 3e). Corroborating with this, SV density was the lowest in gene bodies and increased with distance in flanking regions (Fig. 3f), suggesting that SVs affecting regulatory sequences are likely to be under less stringent selection pressure than those disrupting encoding sequences. Besides, we found that 75% of all SVs overlapped with TEs (Supplementary Fig. 5c). We further identified 'SV gene', being the closest gene to the given SV within a 10-kb radius. In total, we determined 11,377 SV genes based on the syntenic pan-genome, including 9,442 expressed genes. These expressed SV genes were then separated into six groups based on the distance between SVs and corresponding genes (Fig. 4a). The 27 *B. oleracea* genomes were separated into two groups (presence and absence) based on the SV genotype of each SV gene. To be independent of the reference genome used for SV calling, we defined the genotype with more sequence as 'presence' and the genotype with less sequence as 'absence'. Comparison of SV gene expression between absence and presence groups revealed high percentages of SVs that have an effect on gene expression, decreasing with distance from 83% when located in the CDS region to 66% when located in 5–10 kb upstream of SV genes (Fig. 4a and Supplementary Table 13). In total, for 69% (6,526) of the 9,442 SV genes, the SV was associated with gene expression changes. Of these 6,526 SV genes, SV presence was associated with significantly ($P = 1.48 \times 10^{-11}$, binomial test) more SV genes with suppressed expression (3,536 SV genes) than promoted expression (2,990 SV genes; Fig. 4b).

We also found that methylation was strongly associated with the suppressed expression of SV genes (Supplementary Note 4 and Supplementary Fig. 8a). We examined the sequence signature of the SV presence genotype for the 3,536 suppression SVs and found that their CpG site density was significantly higher than that of the 2,990 promotion SVs (Fig. 4c). The methylation levels of these suppression SVs were also significantly higher than that of the promotion SVs (Supplementary Fig. 8b). Both the increased density of CpG sites and

their increased methylation levels resulted in a strong increase of highly methylated CpG islands in suppression SVs compared to promotion SVs (Fig. 4c). Besides suppression SVs, promotion SVs were identified that were associated with increased expression of SV genes. We investigated the sequence composition of promotion SVs and found significant ($P < 0.001$, permutation test) enrichment of transcription factor (TF)-binding sites, including TCP, MYB, NAC, ERF and GRAS (Supplementary Table 14). These specific domains, together with low sequence methylation levels and few CpG islands in promotion SVs, may cause increased transcription of corresponding SV genes.

To further assess the strength of the effect of SVs on gene expression in *B. oleracea* genomes, we calculated the mean expression of corresponding SV genes for each of the two genotype groups (Fig. 4b). SVs affected gene expression ranging from over tenfold reductions to over tenfold increases, with most expression changes falling between one-third and three times (Fig. 4b,d). Furthermore, SVs that affect gene expression were enriched within 3 kb flanking regions of genes. These results indicate the important role of SVs in fine-tuning gene expression levels.

We then used the nonredundant 50,153 SVs to construct an integrated graph-based genome with the T10 genome as a standard linear base reference. By mapping reads of 704 *B. oleracea* accessions to this graph-based genome, we revealed a total of 40,028 SVs in the population (Supplementary Note 4). We randomly selected 62 SVs, of which 59 were validated by PCR amplification (Supplementary Fig. 9 and Supplementary Table 15). Besides SVs, we identified 4,901,625 SNPs and 573,033 InDels in the population. Linkage disequilibrium (LD) analysis between these SVs and SNPs showed that 54.78% of SVs had weak LD ($r^2 < 0.5$) with SNPs (Supplementary Fig. 10), indicating that SVs cannot be fully represented by SNPs in this genomic study. Of the 7,685 SV genes found in the *B. oleracea* population, 4,366 SV genes were expressed and 3,216 SV genes were used for downstream analysis (Methods). The percentage of SVs significantly ($P < 0.05$) associated with the expression of SV genes ranged from 68% in the gene body to 59% 5–10 kb away from the genes. In total, 61% of these SVs were substantially associated with expression changes of their SV genes, slightly less than 69% among the 27-genome assemblies. The SV presence was substantially associated with suppressed expression of 1,071 (55%) genes or promoted expression of 888 (45%) genes, similar to that of the 27-genome analysis (54% suppression, 46% promotion).

We also performed SV-based eGWAS analysis using 17,696 expressed genes and 40,028 SVs as traits and markers, respectively (Methods). The expression of 8,180 genes was significantly associated ($P < 1.00 \times 10^{-10}$) with at least one SV. In total, 50,076 SV signals were identified, among which 23% (11,536) and 77% (38,540) were intrachromosomal and interchromosomal signals, respectively (Supplementary Table 16). Of the 11,536 intrachromosomal SV signals, 1,335 were *cis*-regulatory SVs, with 49% and 51% of them suppressing and promoting gene expression, respectively. The remaining 48,741 SV signals were *trans*-regulatory SVs, with 47% and 53% suppressing and promoting gene expression, respectively. These results further indicate the important and complex regulatory role of SVs in gene expression.

## Expression alterations by SVs associated with morphotypes

We adopted the case–control GWAS strategy[30,31] to identify SVs associated with different morphotypes of *B. oleracea* (Methods). Using the cauliflower/broccoli accessions characterized by large arrested inflorescences as the case group, we obtained 1,655 SV signals with $P < 8.16 \times 10^{-45}$, representing the top 5% signals (Fig. 5a). These SVs were assigned to 492 SV genes (SV in gene bodies or 3 kb flanking regions), of which 378 were expressed, harboring 122 suppression and 109 promotion SVs. One suppression SV ($P = 1.54 \times 10^{-108}$; 112 bp) was located 643 bp upstream of the translation start site of the gene *BoPNY* (PENNYWISE; Fig. 5b), which functions in maintaining inflorescence meristem

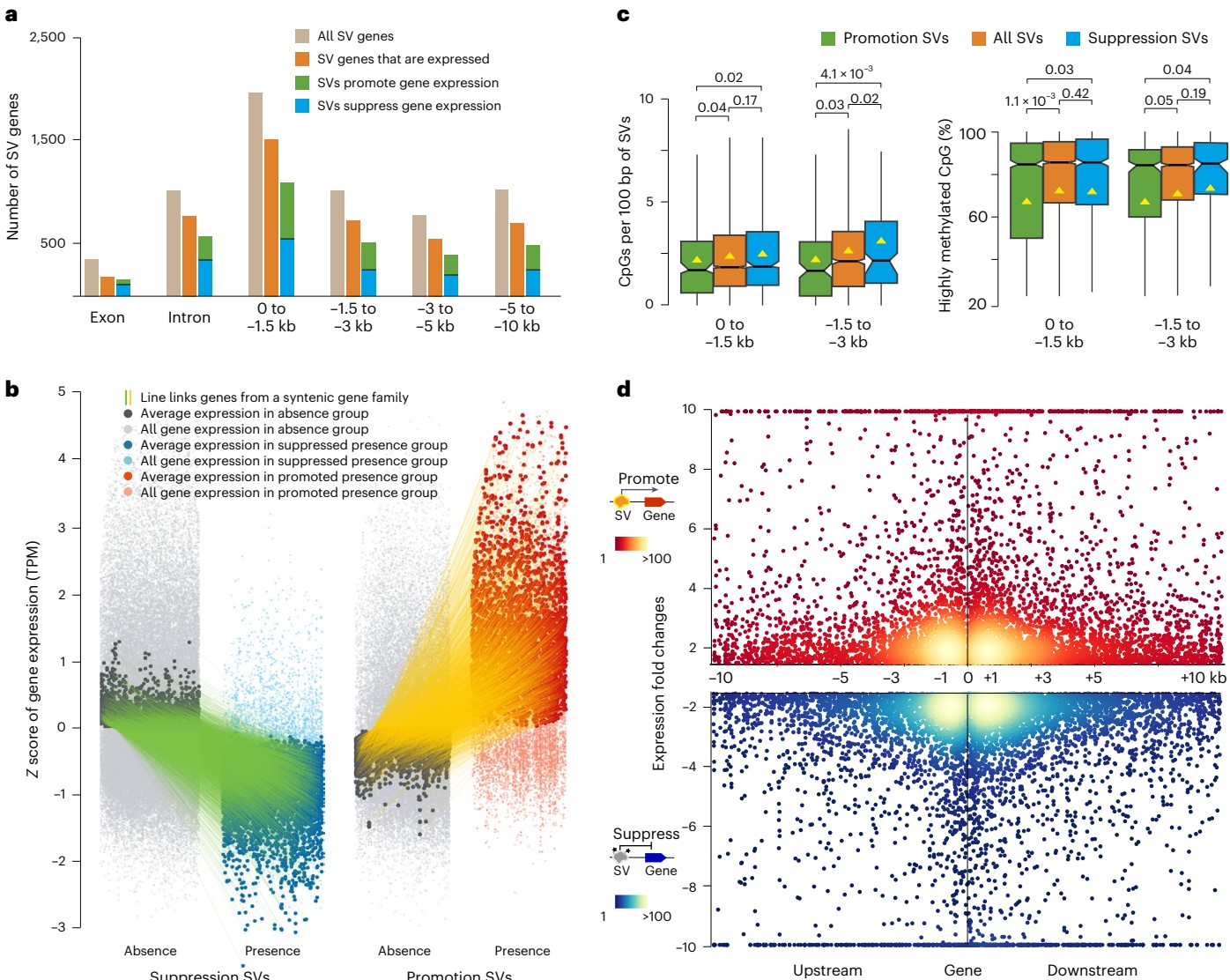

**Fig. 4 | SVs identified by multiple genome comparisons and their effect on the expression of associated genes (SV genes). a,** Different types of SV genes, based on the location of the SV relative to the gene, with data on expression, show a high proportion of SV genes with gene expression changes. **b,** The expression of SV genes from 6,526 syntenic gene families, with separated expression values for the absence and presence genotype groups (of corresponding SV). The $x$ axis shows two groups, of which 3,536 and 2,990 syntenic gene families associated with suppression and promotion SVs, respectively. The $y$ axis shows the normalized ($z$ score) expression values. The green/yellow lines link the average expression values from each syntenic gene family for their presence and absence of genotype groups. **c,** Comparison of CpG island density and the ratio of highly methylated CpG islands between different types of SVs in −1.5 kb ($n$ = 484 versus 369 versus 2,794; permutation test for 10,000 times; centerline, median; triangle, mean; box limits, first and third quartiles; whiskers, 1.5× IQR) or −3 kb ($n$ = 153 versus 148 versus 1,391; permutation test for 10,000 times; centerline, median; triangle, mean; box limits, first and third quartiles; whiskers, 1.5× IQR). **d,** The expression fold changes of SV genes between the presence and absence of genotype groups. The black stars below the term 'Suppress' denote DNA methylation modifications. The $x$ axis shows the distance between SV and SV genes.

identity and floral whorl morphogenesis[32]. This SV was under strong negative selection in the arrested inflorescence morphotype, being present in 2% (4 of 195) of cauliflower/broccoli accessions, contrasting to a presence of 89% (386 of 434) of control group accessions (Fig. 5c). More importantly, *BoPNY* was significantly higher expressed ($P$ = 3.00 × 10⁻³) in the absence genotypes (the major allele in cauliflower/broccoli) than in the presence genotypes (Fig. 5d). The methylation levels of both the presence SV and its flanking sequences were significantly ($P$ = 8.55 × 10⁻⁶) higher than that of the absence genotype, which was negatively associated with the transcription level of *BoPNY* (Fig. 5e). We also identified two promotion SVs located closest to gene *BoCKX3*. Cytokinin oxidase (CKX) catalyzes the degradation of cytokinin and thus negatively regulates cell proliferation of plants[33]. Mutants of *ckx3* and its ortholog *ckx5* form more cells and organs become larger[34].

One SV (SV1; $P$ = 5.81 × 10⁻¹⁶²) involved a 316-bp Helitron-type TE insertion located 86 bp downstream of the translation stop site of *BoCKX3* (Fig. 5f). SV1 was present in 97% (208 of 214) of the cauliflower/broccoli accessions, contrasting to only 0.2% (1 of 431) of accessions in the control group (Fig. 5g). The other SV (SV2, 257 bp) was located in last exon of *BoCKX3*, resulting in a frame-shift mutation. SV2 was present in only 0.5% (1 of 213) of cauliflower/broccoli accessions, compared to 29% (126 of 434) of accessions in the control group (Fig. 5f,g). These two SVs form four potential haplotypes of *BoCKX3*; however, the haplotype containing two SVs does not exist in our *B. oleracea* population (Fig. 5h). The expression of *BoCKX3* in haplotype 3 was significantly higher than in haplotypes 1 and 2 (Fig. 5i), supporting the expression-promoting effect of this downstream SV1. *BoCKX3* was highly expressed in leaves but not in other organs such as the curd during curd development in

cauliflower/broccoli (Fig. 5j). One hypothesis is that *BoCKX3* negatively regulates leaf growth, thus saving energy for fast proliferating of curds. These examples demonstrate the bidirectional impacts of SVs on gene expression, specifically associated with morphotypes of cauliflower/broccoli.

GWAS analysis was also performed using cabbage accessions as the case group, characterized by the leafy heads (Supplementary Note 5 and Supplementary Figs. 11 and 12). We revealed two promotion SVs (SV1 and SV2) located closest to *BoKAN1*, which regulates leaf adaxial/abaxial polarity[35–37]. SV1 was introduced by a 970-bp TE (PIF/Harbinger) insertion, which was under strong negative selection in cabbage accessions (Supplementary Fig. 11b and c), and SV2 was introduced by a 157-bp TE (Helitron) insertion, which was also under negative selection in cabbage accessions. Among the four haplotypes formed by the two SVs (Supplementary Fig. 11d), *BoKAN1* was significantly ($P = 3.60 \times 10^{-7}$) lower expressed in haplotypes 1 and 2 that lacked SV1 than in haplotypes 3 and 4 that harbored SV1 (Supplementary Fig. 11e). We also revealed one promotion SV ($P = 3.69 \times 10^{-91}$) located closest to *BoACS4* (Supplementary Fig. 12a), which encodes the key regulatory enzyme involved in the biosynthesis of the plant hormone ethylene[38,39]. This insertion was under strong negative selection in cabbage (Supplementary Fig. 12b). Expression of *BoACS4* in cabbage accessions lacking this insertion was significantly lower ($P = 1.90 \times 10^{-14}$) than in control group accessions harboring the insertion (Supplementary Fig. 12c).

Another interesting SV was present in all 18 ornamental kale accessions, but absent in any other accession. This SV was a 280-bp TE (PIF/Harbinger) insertion, located 289 bp upstream of the translation start site of a MYB TF (hereafter referred to as *BoMYBtf*; Fig. 6a and b). Previously, MYB TFs were found to be associated with purple traits in cultivars of *B. oleracea*, such as kale, kohlrabi and cabbage[40]. The expression level of *BoMYBtf* was significantly higher in ornamental kale than in other morphotypes (Fig. 6c), indicating that this TE insertion was associated with the promoted expression of *BoMYBtf*. TF-binding sites (that is NAC, TCP and ERF), which were substantially enriched in promotion SVs as aforementioned, were also found in this PIF/Harbinger TE sequence (Fig. 6d). We hypothesize that these TF-binding sites, hitchhiking with the TE insertion, are causal factors promoting the transcriptional activity of *BoMYBtf*.

The role of this PIF/Harbinger TE in increasing transcription of *BoMYBtf* in ornamental kale was further validated by the luciferase reporter experiment (Fig. 6e). Briefly, the MYB promoters of ornamental kale T18 (with TE), wild *B. oleracea* T10 (without TE), cabbage JZS T20 (without TE) and the SV (TE itself) were fused in pMini-LUC as reporters and transfected into tobacco leaves (Methods). The LUC/REN ratio of mini-T18 and mini-SV was significantly higher ($P < 0.05$) than that of other samples, while no significant difference was observed between mock, mini-T10 and mini-JZS, confirming the expression promotion effect of this PIF/Harbinger TE. Moreover, we investigated this PIF/Harbinger TE across all the 27 *B. oleracea* genomes. We found 60 insertions located within 3 kb flanking regions of genes, with 44 associated genes being expressed (Fig. 6f). When comparing their

expression among the 27 genomes, 31 genes harboring the insertion showed higher expression levels than their counterparts lacking the insertion, whereas this insertion in the remaining 13 genes did not result in increased expression (Fig. 6g). These results further support the common transcription promotion function of this PIF/Harbinger TE insertion in *B. oleracea* genomes.

## Discussion

Different highly diverse morphotypes have evolved in *B. oleracea*. To explore the genomic basis underlying the evolution of these diverse morphotypes, we generated chromosome-level genome assemblies of representative *B. oleracea* accessions and constructed a high-standard *B. oleracea* pan-genome from 27 genomes. We revealed patterns of differential gene loss associated with specific morphotypes of *B. oleracea*. More importantly, using the pan-genome, together with multi-omics datasets from large-scale populations, we systematically identified SVs in the *B. oleracea* population and showed that SVs exert bidirectional effects on the expression of numerous genes. Notably, many SVs affecting gene expression were under strong selection in specific morphotypes of *B. oleracea*.

There are two groups of SV genes as follows: one in which the presence of genotype suppresses gene expression and another in which the presence of genotype promotes gene expression. Previous studies showed that TEs were always highly methylated, simultaneously suppressing their transposition activity and silencing the expression of adjacent genes[41]. As most SVs overlap with TEs and are likely introduced by TEs, the TE methylation mechanism may be the main causal factor for the suppressive effects of SV genes. Our whole-genome methylome analysis supported this, showing that the suppression SVs were associated with higher levels of sequence methylation around genes. Meanwhile, TF-binding elements were found to be enriched in promotion SVs. Some of these binding sites were introduced through the retention of fractionated TE sequences, which is supported by frequent cases of promotion SVs annotated as TEs. For promotion SVs that do not overlap with TEs, the TF-binding sites are likely part of the original promoters of these genes, such that the absence of genotype results in downregulated expression of its SV gene.

In GWAS analysis using SVs as markers, we identified strong signals associated with different morphotypes. These SVs affected the expression of important genes associated with specific morphotypes. The SV gene examples provided in this study serve as solid illustrations for the continuous occurrence of both transcriptional suppressing and promoting abilities of SVs in *B. oleracea*. These results underscore the crucial role of SVs in fine-tuning gene expression dosage, acting as an efficient natural mutagenic factor analog to a dosage knob that turns down or up the expression of corresponding SV genes. Hereby SVs emerge as pivotal contributors to the domestication and morphotype diversification of *B. oleracea*. In addition to this, the current study does not exclude other mechanisms, like SVs affecting protein-coding sequences of genes, as well as SNPs and InDels.

**Fig. 5 | GWAS analysis identified SVs associated with the cauliflower/ broccoli morphotype and details of SVs that change the expression of genes *BoPNY* and *BoCKX3*. a**, Manhattan plot showing the SV signals associated with cauliflower/broccoli (significance was calculated by two-tailed Fisher's exact test. A Bonferroni-corrected $P < 0.05$ was interpreted as significant). The light red dots show the top 5% $P$ values and deep red dots show the top 1% $P$ values. **b**, One SV is associated with *BoPNY*. **c**, The number of accessions with presence or absence SV (associated with *BoPNY*) genotype for broccoli/cauliflower accessions and all the other accessions (statistical test: two-tailed Fisher's exact test). **d**, Expression comparison of *BoPNY* between SV presence and absence accessions (two-sided Student's *t* test; centerline, median; box limits, first and third quartiles; whiskers, 1.5× IQR). **e**, Sequence methylation level around *BoPNY* between absence and presence genotype groups, which is negatively associated

with the expression level of the gene. **f**, Two SVs associated with *BoCKX3*. **g**, The number of accessions with presence or absence SV (associated with *BoCKX3*) genotypes for broccoli/cauliflower accessions and all other accessions (statistical test: two-tailed Fisher's exact test). **h**, The four possible haplotype groups are formed by two SVs. Haplotype 4 was not detected in our population. **i**, Expression comparison of *BoCKX3* between the three haplotype groups (two-sided Student's *t* test; centerline, median; box limits, first and third quartiles; whiskers, 1.5× IQR). **j**, Expression of *BoCKX3* in different tissues of cauliflower and cabbage, highlighting high expression of this gene in leaf 2 of cauliflower. Leaf 1 denotes fresh leaf before curd initiation; leaf 2 denotes fresh leaf during curd development; curd 1 denotes developing curd; curd 2 denotes mature curd. 'N' indicates a missing value as cabbage makes no curds.

In summary, the high-quality genome assemblies, pan-genome and graph-based SV characterization in *B. oleracea*, along with findings on large-scale gene expression variation introduced by SVs associated with specific morphotypes, provide a comprehensive landscape of genomic, genetic and transcriptional variations for this species. These results enhance our understanding of the mechanism underlying

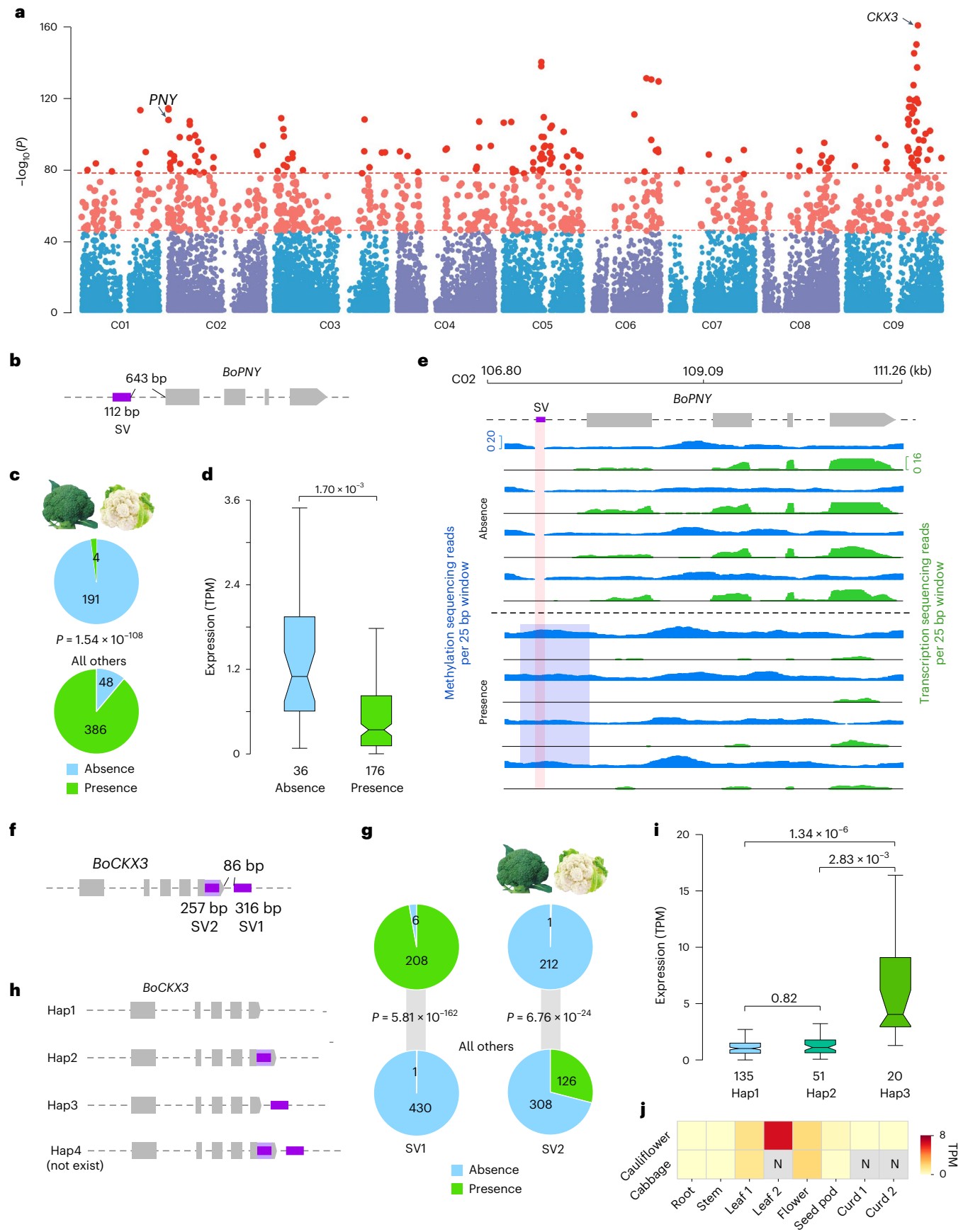

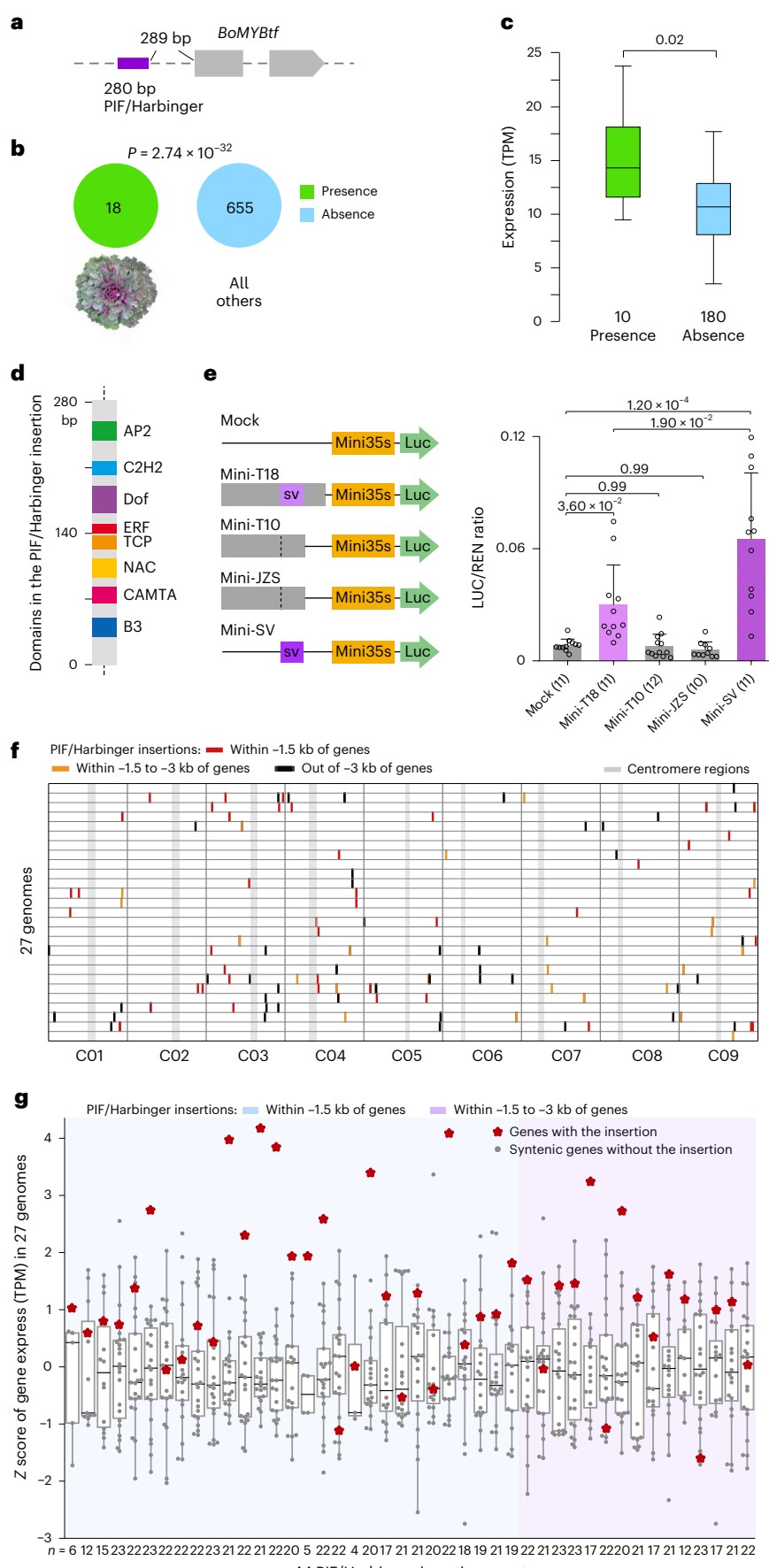

**Fig. 6 | SVs derived by PIF/Harbinger-type TE insertions promote the expression of *BoMYBtf* in ornamental kale; overview of increased expression levels of genes in the 27 *B. oleracea* accessions with PIF/Harbinger insertions in their promoter regions. a**, One SV (PIF/Harbinger-type TE insertion) is associated with *BoMYBtf*. **b**, The number of accessions with presence or absence of SV (associated with *BoMYBtf*) genotypes for ornamental kale accessions and all other accessions (statistical test: two-tailed Fisher's exact test). **c**, Expression comparison of *BoMYBtf* between SV presence and absence accessions (two-sided Student's *t* test; centerline, median; box limits, first and third quartiles; whiskers, 1.5× IQR). **d**, TF-binding elements identified in the PIF/Harbinger insertion. **e**, Schematic diagrams of reporter constructs used for the LUC/REN assay. The upstream sequences of *BoMYBtf* from ornamental kale T18 (with TE, 1,239 bp), wild *B. oleracea*

T10 (without TE, 951 bp), cabbage T20 (without TE, 968 bp) and the SV sequence (TE itself, 280 bp). The empty vector was set as mock control. The activities of these promoter constructs are reflected by the LUC/REN ratio (two-sided Student's *t* test; data are presented as the mean ± s.d.). **f**, Distribution of the PIF/Harbinger insertion in the 27 *B. oleracea* genomes. **g**, Boxplot showing normalized (*z* score) expression of 44 syntenic gene families, with a PIF/Harbinger insertion within a −3 kb region from the nearest genes. The light blue and light purple backgrounds denote these syntenic gene families with PIF/Harbinger insertions located within −1.5 kb and −3 kb to −1.5 kb, respectively, of corresponding gene members (red stars); whereas the gray dots denote their syntenic gene members without PIF/Harbinger insertion (centerline, median; box limits, first and third quartiles; whiskers, 1.5× IQR).

the rapid evolution and domestication of different vegetable crops in *B. oleracea*, highlighting the important role of SVs herein through modulating gene expression. Our findings illustrate the importance of SV-associated fine-tuning of gene expression in future crop breeding programs.

## Online content

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

## Methods

### Plant materials and sequencing

We collected genome resequencing data of 704 *B. oleracea* accessions, including 36 wild *B. oleracea*, 310 heading cabbage, 153 cauliflower, 63 broccoli, 46 kohlrabi, 21 curly kale, 18 ornamental kale, 24 Chinese kale, 20 brussels sprout, seven Tronchuda kale and six collard green accessions (Supplementary Tables 1 and 2). Among these plant materials, 415 accessions are generated in this study, and 289 accessions are obtained from two previous studies[4,5]. Genomic DNA from young leaves of the 415 accessions was extracted. DNA libraries were constructed following the manufacturer's instructions and sequenced on the Illumina HiSeq 2000 platform, resulting in an average of 22.69× coverage reads per accession (Supplementary Table 1).

A total of 22 *B. oleracea* accessions were selected for de novo genome assembly in this study, among which 17 accessions were sequenced by the Institute of Vegetables and Flowers, Chinese Academy of Agricultural Sciences (IVF-CAAS) and five accessions (T06, T24, T25, T26 and T27) were sequenced by Plant Breeding, Wageningen University and Research (PBR-WUR). Methods for library construction and sequencing (PacBio, Nanopore, Bionano, Hi-C, Illumina, mRNA-Seq and methylation data) of plant materials are provided in the Supplementary Note 1.

### Reads mapping and variant calling

Raw resequencing reads of 704 *B. oleracea* accessions were filtered using Trimmomatic (v0.39)[42]. Reads containing adapters, duplicated reads and low-quality reads (containing >5% unknown bases or average base quality <20) were removed. Clean reads were mapped to the JZS v2.0 (ref. 22) reference genome using BWA-MEM (v0.7.17-r1188)[43] with default settings. SAMtools (v1.9)[44] was used to transform the SAM into BAM files and to sort the BAM files.

Variant calling of all 704 accessions was performed following the best practices of the Genome Analysis Toolkit (GATK; v4.1.4.0)[45] with default parameters. CreateSequenceDictionary was first used to build an index of the JZS v2.0 (ref.22) reference genome. After that, MarkDuplicates was used to mark duplicated reads in each sample. Next, HaplotypeCaller was used to produce GVCF files on a per-sample basis, following which CombineGVCFs was used to merge per-sample GVCF files of the 704 accessions into a single GVCF file. Then, GenotypeGVCFs was used to perform joint genotyping to identify SNPs and InDels. Finally, variations were filtered with parameters '--mac5, --minDP5, --minQ30, --max-missing 0.8, --maf 0.02' using VCFtools (v0.1.6)[46]. These filtered variations were further annotated using SnpEff (v4.3)[47].

### De novo genome assembly

Jellyfish (v2.2.10)[48] and GenomeScope (v2.0)[49] were used to estimate the genome size for each of the 22 newly sequenced accessions using Illumina reads. NextDenovo v2.2 (https://github.com/Nextomics/Next-Denovo) was used for de novo genome assembly using PacBio or Nanopore long-reads with default parameters for the 17 IVF-CAAS accessions. The resulting contigs were polished using both long- and short-reads by Nextpolish (v1.1.0)[50] with default parameters. For seven genomes with relatively high level of heterozygosity, minimap2 (v2.18-r1015)[51] was used to map long-reads to each of the assemblies, following which purge_dups (v1.2.3)[52] was used to remove falsely duplicated regions in the primary assemblies. For the five PBR-WUR accessions, SMARTdenovo (v1.0)[53] was used to assemble each of the genomes with parameters '-c 1 -k 17'. Assembled contigs were then polished using Nanopore reads for two iterations, followed by Illumina reads for three iterations. For Nanopore reads polishing, minimap2 (v2.18-r1015)[51] was used to map raw Nanopore reads to raw SMARTdenovo assembly or polished assembly after the first round with parameter '-x map-on'. The resulting file was submitted to Racon (v1.3.3) for sequence polishing using default parameters[54]. For Illumina reads polishing, Illumina paired-end reads were aligned to polished contigs from the previous iteration

using BWA-MEM (v0.7.17-r1188). The resulting bam file was sorted by SAMtools (v1.9)[44] and then subjected to Pilon (v1.23)[55] with default parameters for assembly improvement. Assembly completeness was evaluated using BUSCO[56] based on 1,614 single-copy orthologous genes of the Embryophyta dataset v10.

### Construction of pseudomolecules

For 16 of 17 IVF-CAAS accessions, Hi-C reads for each genome were aligned to the corresponding contigs using Juicer (1.9.9)[57]. The 3D DNA (v180922)[58] was used to correct the potential mistakes and to order, orient and scaffold the sequences. The generated scaffolds were then reoriented to produce chromosome-level assemblies using ALLHiC (v0.9.8)[59]. Then, Juicebox (v1.9.8)[60] was used to visualize and interactively (re)assemble the genome by manually adjusting chromosome boundaries and correcting the misassembles. Finally, the order of pseudochromosomes was verified by whole-genome alignment between each of our genomes and the JZS v2.0 reference genome[22] using nucmer (v4.0.0)[27]. We anchored contigs of T21 into pseudochromosomes by mapping to JZS v2.0 reference genome[22]. For the five PBR-WUR accessions, the generated optical mapping molecules were de novo assembled into genome maps using Bionano Solve Pipeline (v3.4.1) and Bionano Access (v1.3). 'HybridScaffold' module in Bionano Solve Pipeline was then used to perform hybrid scaffolding between polished contig sequences and Bionano genome maps. As a default parameter, the hybrid scaffolding pipeline did not fuse overlapped ONT contigs, which were indicated by the optical maps, but added a 13-bp gap between the two contigs. We checked all 13 bp gaps and aligned both 50 kb flanking regions with BLAT (v36)[61]. The two flanking contigs were joined if one alignment was detected[23]. To construct chromosome-level pseudomolecules, we mapped super-scaffolds to the HDEM reference genome[23,62].

### Transposable element annotation

RepeatModeler (v2.0.1)[63] was used to construct a nonredundant TE library with default parameters. LTR_Finder (v1.07)[64] with default parameters and LTR_harvest (v1.6.1)[65] with parameters '-similar 30 -seed 20 -minlenltr 100 -maxlenltr 3500 -motif TGCA' were used to construct LTR-RT libraries. LTR_retriever (v2.9.0)[66] was used to merge the results of LTR_Finder and LTR_harvest and to generate a nonredundant LTR-RT library. Thereafter, we combined the LTR-RT library and the TE library, the redundancy of which was removed using CD-HIT (v4.8.1)[67] with parameters '-c 0.8 -aS 0.8'. Finally, genome-wide repetitive sequences were annotated and classified based on the constructed library using RepeatMasker (v4.1.0; http://repeatmasker.org) with default parameters. Full-length LTR-RTs identified by LTR_retriever were clustered by CD-HIT (v4.8.1) with parameters '-c 0.9 -aS 0.9'. The insertion time of the intact LTR-RT was calculated using the base substitution rate of $1.3 \times 10^{-8}$ per site per year.

### Gene prediction and functional annotation

Protein-coding gene models were predicted based on repeat-masked assemblies using a strategy that combined homology-based, transcripts-based and ab initio predictions. For homology-based gene prediction, exonerate (https://github.com/nathanweeks/exonerate) was used to detect homologous gene models with default parameters. For transcripts-based prediction, Trinity (v2.8.5)[68] was used to assemble mRNA-seq reads into transcripts, which were subsequently subject to PASA (v2.4.1)[69] for gene model prediction. For ab initio prediction, AUGUSTUS (v3.2.3; https://github.com/Gaius-Augustus/Augustus) and GeneMark (v4.69_lic)[70] were used to predict gene structures, incorporating transcriptome data as evidence. Finally, EVidenceModeler (v1.1.1)[71] was used to merge gene predictions from the three approaches and generate a weighted consensus gene set for each genome assembly. Predicted gene models were checked to ensure the correct placement of start and stop codons. Genes containing internal stop codons or

lacking the start/stop codons were removed. BUSCO was used to evaluate the completeness of gene annotation. InterProScan (v5.46-81.0)[72] was then used to predict motifs and functional domains. Gene ontology (GO) information was extracted from the output of InterProScan. GO enrichment analysis was performed by ClusterProfiler (v4.0.5)[73].

## Phylogenetic analysis

For the 704 resequencing accessions, a total of 6,704,072 filtered SNPs were used for phylogenetic analysis by FastTree (v2.1.11) with default parameters. The online tool iTOL (http://itol.embl.de) was used to visualize the constructed tree. We also constructed a phylogenetic tree, including the 27 de novo assembled *B. oleracea* genomes and the *Arabidopsis* genome (outgroup). Single-copy genes between these 28 genomes were determined by OrthoFinder (v2.4.0)[26] with default parameters. The coding sequences of the single-copy gene families were aligned using MUSCLE (v3.8.1551)[74]. Gblock (v0.91b)[75] was used to extract the conserved sequences among the 28 genomes. Seqkit (v2.1.0)[76] was used to concatenate sequences for phylogenetic analysis. The phylogenetic tree was constructed using FastTree[77] with default parameters and visualized using iTOL.

## TAD structure prediction

TADs of *B. oleracea* T10 and *B. rapa* A03 (ref. 78) genomes were predicted using FAN-C software (v0.9.24)[79]. Hi-C reads of each accession were mapped to the corresponding genome to obtain fragment-level Hi-C object at various bin sizes using 'fanc auto' function with default parameters. The 'fanc insulation' function was then used to calculate insulation scores, and 'fancplot' function was further used to plot insulation scores in 500 kb windows.

## Identification of orthologous and syntenic gene family

OrthoFinder (v2.4.0)[26] was used to identify orthologous gene families among 27 *B. oleracea* genomes with default parameters. The tool mSynOrths (v0.1; https://gitee.com/zhanglingkui/msynorths) was used to identify syntenic gene pairs of 27 genomes with parameters '-n 20 -m 0.6'. Genes that had no syntenic pairs with all other genomes and no tandem duplicates were defined as orphan genes, which were excluded in syntenic gene family analysis. Orthologous and syntenic core, softcore, dispensable and private gene families were defined as those that were present in all 27 accessions, in 25–26 (>90%) accessions, in 2–24 accessions and in only one accession, respectively.

## Homoeologous gene identification and retention analysis

A total of 20,924 genes that were lost in some genomes but retained in >50% of the 27 genomes were selected for syntenic gene retention analysis. Accessions that experienced gene loss were then determined for each of the lost genes. A morphotype specifically lost gene was defined if more than 70% of the accessions with gene loss occurred in the given morphotype.

Three subgenomes (LF, MF1 and MF2) of each of the 27 *B. oleracea* genomes were constructed using a previously reported method[8]. For homoeologous gene retention variation analysis, three-copy homoeologs in the wild *B. oleracea* genome T10 were treated as ancestral three-copy genes. A total of 3,755 three-copy genes in T10 that lost part of copies in some genomes but remained all three copies in >50% of the 27 genomes (>50%) were used to investigate CNV of homoeologs among *B. oleracea* morphotypes.

## SV identification

Whole-genome alignments between each of the 26 *B. oleracea* genomes and T10 reference genome were performed using nucmer (v4.0.0) with parameters '-maxmatch -c 100 -l 50'. We then filtered the alignments using delta filters with parameters '-m -i 90 -l 100'. The filtered delta files were converted into coords files using show-coords with parameters '-T -H -r -d'. Thereafter, SyRI (v1.5.4)[28] was used to detect inversions and

translocations with default parameters. For PAVs, the output of INS and CPG variations from SyRI was defined as presence variations, and that of DEL and CPL variations was defined as absence variations. The same type of variations with continuous (or overlapped) coordinates on the reference genome was merged as a single SV. Circos (v0.69-8)[80] was used to visualize the distribution of these SVs.

## SV validation

Hi-C data were used to validate randomly selected SVs longer than 8 kb. We mapped Hi-C paired-end reads to the corresponding genome assemblies and manually checked the interaction heatmap for the regions containing SVs. For the randomly selected SVs shorter than 8 kb, we mapped long-reads to the corresponding genome assemblies and manually checked the alignments at the boundaries of these SVs. In addition, we randomly selected five SVs and performed PCR amplification to examine the fidelity in 20 samples that were randomly selected from the resequenced accessions.

## SV genes and associated expression analysis in assemblies

We assigned each SV to its closest gene that was located within 10 kb of flanking regions of the given SV. These genes were referred to as SV genes. Syntenic genes among the 27 genomes were further classified into two groups (the genotype with more sequence as 'presence' and the genotype with less sequence as 'absence'), allowing only one SV occurring in 10 kb flanking regions of the SV gene. Among these, syntenic genes that had different SV genotypes in at least four genomes were selected for gene expression analysis. mRNA-seq data of 22 genomes were used to quantify gene expression for the two genotype groups. An SV gene was considered expressed if more than 60% of samples in the given group showed a TPM value of ≥1. The mean TPM value was used to compare gene expression levels between the two genotype groups. Promoting SVs were defined if the mean TPM of syntenic gene with SV presence was at least 1.5-fold higher than that of SV absence. Similarly, suppressing SVs were defined if the mean TPM of syntenic gene with SV absence was at least 1.5-fold higher than that of the SV presence.

## Prediction of TF-binding sites

TF-binding sites of SV sequences were predicted using the online tool PlantTFDB (http://planttfdb.gao-lab.org/)[81]. A permutation test of 1,000 times was used to evaluate TF-binding site number differences between promoting SVs and suppressing SVs.

## Methylation analysis

Whole-genome bisulfite sequencing reads of 16 accessions were mapped to their corresponding genomes using Bismark v0.20.0 (ref. 82). The CpG methylation profile was analyzed in this study. Methylation ratio of each cytosine covered by at least three reads was calculated by dividing the number of methylated CpG reads by the total number of CpG reads. The methylation level of SV sequences was calculated using a weighted method[83]. In brief, the methylation level of an SV sequence was calculated by the total methylated CpG reads (≥3) divided by the total CpG reads.

## Graph-based genome construction

PAV sequences were used to construct a graph-based genome. We used CD-HIT (v4.8.1) to cluster and remove redundant PAV sequences of the 27 genomes with parameters '-c 0.95 -n 10 -aS 0.95 -M 0 -T 0'. In each cluster, one PAV was randomly selected as a representative to construct the graph-based genome using vg toolkit (v1.33.0)[84], with T10 being the based linear genome. Vg index was used to store the graph in the xg and gcsa index pair with default parameters. To genotype SVs in 704 *B. oleracea* accessions, we mapped Illumina short-reads from each accession to the indexed graph-based genome using vg map with default parameters. Low-quality alignments were excluded using vg

pack with parameter '-Q 5'. SV genotyping of each accession was then performed using vg call with parameters '-a -s'. Genotyped SVs with less than three supporting reads were marked as 'missing'. Finally, the genotyped SVs of 704 accessions were merged into one vcf file using bcftools merge (v1.13)[85] with parameter '-m'.

## Case–control GWAS analysis

We adopted the case–control GWAS strategy, which was widely used in disease gene mapping for humans[30,31], to identify SVs that were substantially associated with different morphotypes of *B. oleracea*. Briefly, a GWAS analysis was performed between the case group (individuals belonging to a specific morphotype) and the control group (individuals belonging to all the other morphotypes). Significance was tested by a two-tailed Fisher's exact test and adjusted by Bonferroni correction.

## SV genes and associated expression analysis in the population

The association analysis between SV and gene expression in *B. oleracea* population was performed using the 223 accessions with mRNA-seq data. We assigned each SV to its nearest gene based on the T10 reference genome. The gene that is located within 10 kb of the flanking region of a given SV was defined as an SV gene, similar to those described in the SV gene analysis using 27 assembled genomes. We filtered the SV locus in the 223 accessions using the following criteria: (1) at least 60 samples were successfully genotyped and (2) at least ten samples were present in each of the two genotype groups mentioned above. An SV gene was considered to be expressed in the *B. oleracea* population if it had TPM values of ≥5 in more than 30% of the samples. The significant association ($P < 0.05$) between SV genotype and gene expression was tested using the Mann–Whitney $U$ test[86]. The definition of promoting SVs and suppressing SVs was the same as mentioned above.

## eGWAS analysis

We performed SV-based eGWAS analysis using the 223 accessions with mRNA-seq data. Expressed genes with more than 10% of all samples showing TPM ≥ 10 were used as traits, and 40,028 SVs were used as markers. The significant association between SV and gene expression was analyzed by the software GEMMA (v0.98.3)[87]. A strict $P$-value threshold ($P < 1.00 \times 10^{-10}$) was used to correct for multiple statistical tests. Significant SV signals located within 20 kb flanking regions of the associated genes were defined as *cis*-regulatory SVs. Other significant signals were defined as *trans*-regulatory SVs.

## Luciferase report experiment

For luciferase (LUC)/Renillia luciferase (REN) assays (LUC/REN ratio), different reporter sequences were cloned into pMini-LUC, and the constructions were then introduced into *Agrobacterium tumefaciens* strain GV3101 (pSoup-p19) as reporters. The reporters were transiently expressed in tobacco leaves for 28 d using *Agrobacterium*-mediated transformation. The GLOMAX 20/20 reader was used to detect Firefly and Renilla luciferase activity using the Bio-Lite Luciferase Assay System (Vazyme). Primers used are provided in the Supplementary Note 1.

## Reporting summary

Further information on research design is available in the Nature Portfolio Reporting Summary linked to this article.

## Data availability

All raw sequencing data, assembly and annotation results of *B. oleracea* genomes generated in this study have been deposited in the National Center for Biotechnology Information Sequence Read Archive under the accession number PRJNA1047966 and also in the Genome Sequence Archive at the National Genomics Data Center (https://bigd.big.ac.cn), China National Center for Bioinformation/Beijing Institute of Genomics, Chinese Academy of Sciences, under accession PRJCA017338. Source data are provided with this paper.

## Code availability

Custom scripts and codes used in this study are provided at GitHub (https://github.com/caasivfbioinfo/Bol_pangenome) and Zenodo (https://doi.org/10.5281/zenodo.10202863)[88]. Software and tools used are described in the Methods and Reporting Summary.

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

## Acknowledgements

Through this article, we earnestly honor the memory of Z. Fang (1939–2023), a revered and insightful professor who dedicated his life to the field of vegetable breeding sciences. This study was supported by the National Natural Science Foundation of China (NSFC; grants 31722048 and 31972411 to F.C., 32172578 to Y.Z., 32072570 to H.L. and 32302570 to L.Z.), Central Public-Interest Scientific Institution Basal Research Fund (grant Y2022LM11 to F.C.), Innovation Program of the Chinese Academy of Agricultural Sciences, the China Agriculture Research System of MOF and MARA (CARS-23 to Y.Z.) and the Key Laboratory of Biology and Genetic Improvement of Horticultural Crops, Ministry of Agriculture and Rural Affairs, P.R. China. This study was also funded by the TKI project 'A de novo sequencing catalog of B. oleracea' (KV 1605-004 to G.B.) that was co-supported by two breeding companies (Bejo and ENZA). C.C. was supported by the China Scholarship Council (grant 201809110159).

## Author contributions

F.C., Y.Z. and G.B. conceived and designed the research. Y.W., H.L., J.J., F.H., L.Y., M.Z., J.B. and Y.Z. participated in the material preparation. X.L. and C.C. performed genome assembly. X.L. and K.Z. contributed to Hi-C alignment. X.L. and Y.Y. contributed to the annotation of genome and TEs. X.L. and Lingkui Z. contributed to the identification of syntenic gene family. X.L. performed other bioinformatics analyses. Lei Z., S.C., Q.T. and X.W. performed molecular experiments. F.C. and X.L. wrote the manuscript. F.C., X.L., C.C., G.B., K.Z., Y.W., H.L. and Y.Z. revised the manuscript. All the authors read, edited and approved the manuscript.

## Competing interests

The authors declare no competing interests.

## Additional information

**Correspondence and requests for materials** should be addressed to Kang Zhang, Honghao Lv, Guusje Bonnema, Yangyong Zhang or Feng Cheng.

# Reporting Summary

## Statistics

For all statistical analyses, confirm that the following items are present in the figure legend, table legend, main text, or Methods section.

| n/a | Confirmed | |
|---|---|---|
| ☐ | ☒ | The exact sample size (*n*) for each experimental group/condition, given as a discrete number and unit of measurement |
| ☐ | ☒ | A statement on whether measurements were taken from distinct samples or whether the same sample was measured repeatedly |
| ☐ | ☒ | The statistical test(s) used AND whether they are one- or two-sided *Only common tests should be described solely by name; describe more complex techniques in the Methods section.* |
| ☒ | ☐ | A description of all covariates tested |
| ☐ | ☒ | A description of any assumptions or corrections, such as tests of normality and adjustment for multiple comparisons |
| ☐ | ☒ | A full description of the statistical parameters including central tendency (e.g. means) or other basic estimates (e.g. regression coefficient) AND variation (e.g. standard deviation) or associated estimates of uncertainty (e.g. confidence intervals) |
| ☐ | ☒ | For null hypothesis testing, the test statistic (e.g. *F*, *t*, *r*) with confidence intervals, effect sizes, degrees of freedom and *P* value noted *Give P values as exact values whenever suitable.* |
| ☒ | ☐ | For Bayesian analysis, information on the choice of priors and Markov chain Monte Carlo settings |
| ☒ | ☐ | For hierarchical and complex designs, identification of the appropriate level for tests and full reporting of outcomes |
| ☒ | ☐ | Estimates of effect sizes (e.g. Cohen's *d*, Pearson's *r*), indicating how they were calculated |

*Our web collection on statistics for biologists contains articles on many of the points above.*

## Software and code

Policy information about availability of computer code

| Data collection | Sequencing platforms used to generate the raw data are listed as followed: PacBio SMRT, Oxford Nanopore, Illumina HiSeq, NovaSeq, Bionano Saphyr. |
|---|---|
| Data analysis | The softwares used in this manuscript include Jellyfish v2.2.10, GenomeScope v2.0, Nextpolish v1.1.0, Minimap2 v2.18-r1015, SMARTdenovo v1.0, Racon v1.3.3, purge_dups v1.2.3, Trimmomatic v0.39, BWA v0.7.17-r1188, Samtools v1.9, Pilon v1.23, GATK v4.1.4.0, VCFtools v0.1.6, SnpEff v4.3, NextDenovo v2.2, Juicer 1.9.9, 3D DNA v180922, Circos v0.69-8, ALLHiC v0.9.8, Juicebox v1.9.8, Bionano Solve Pipeline version 3.4.1, Bionano Access version 1.3, BLAT v.36, mSynOrths v0.1, nucmer v4.0.0, RepeatModeler v2.0.1, LTR_Finder v1.07, LTR_harvest v1.6.1,LTR_retriever v2.9.0, CD-HIT v4.8.1, RepeatMasker v4.1.0, Trinity v2.8.5, PASA v2.4.1, AUGUSTUS v3.2.3, GeneMark v4.69_lic, EVidenceModeler v1.1.1, InterProScan v5.46-81.0, ClusterProfiler v4.0.5, OrthoFinder v2.4.0, MUSCLE v3.8.1551, Gblock v0.91b, Seqkit v2.1.0, FastTree v2.1.11, FAN-C v0.9.24, SyRI v1.5.4, Bismark v0.20.0, vg toolkit v1.33.0, bcftools v1.13 and GEMMA v0.98.3. |

For manuscripts utilizing custom algorithms or software that are central to the research but not yet described in published literature, software must be made available to editors and reviewers. We strongly encourage code deposition in a community repository (e.g. GitHub). See the Nature Portfolio guidelines for submitting code & software for further information.

## Data

Policy information about [availability of data](availability of data)

All manuscripts must include a [data availability statement](data availability statement). This statement should provide the following information, where applicable:

- Accession codes, unique identifiers, or web links for publicly available datasets
- A description of any restrictions on data availability
- For clinical datasets or third party data, please ensure that the statement adheres to our [policy](policy)

All raw sequencing data, assembly and annotation results of 22 cabbage genomes generated in this study have been deposited in the Genome Sequence Archive at the National Genomics Data Center (https://bigd.big.ac.cn), China National Center for Bioinformation/Beijing Institute of Genomics, Chinese Academy of Sciences, under accession number PRJCA017338 and also in the National Center for Biotechnology Information Sequence Read Archive under the accession PRJNA1047966.

## Research involving human participants, their data, or biological material

Policy information about studies with [human participants or human data](human participants or human data). See also policy information about [sex, gender (identity/presentation), and sexual orientation](sex, gender) and [race, ethnicity and racism](race, ethnicity and racism).

| | |
|---|---|
| Reporting on sex and gender | N/A |
| Reporting on race, ethnicity, or other socially relevant groupings | N/A |
| Population characteristics | N/A |
| Recruitment | N/A |
| Ethics oversight | N/A |

Note that full information on the approval of the study protocol must also be provided in the manuscript.

# Field-specific reporting

Please select the one below that is the best fit for your research. If you are not sure, read the appropriate sections before making your selection.

☒ Life sciences  ☐ Behavioural & social sciences  ☐ Ecological, evolutionary & environmental sciences

For a reference copy of the document with all sections, see nature.com/documents/nr-reporting-summary-flat.pdf

# Life sciences study design

All studies must disclose on these points even when the disclosure is negative.

| | |
|---|---|
| Sample size | We selected 22 representative accessions including all different Brassica oleracea morphotypes and some wild types for pan-genome construction. The logic of this selection was based on their phylogenetic relationships to ensure their representative of genetic diversity within Brassica oleracea. |
| Data exclusions | No data were excluded from the analyses. |
| Replication | There are 10-15 replicates of Luciferase report experiment for each of 5 different genome sequence independently. All replications were successful and were used. |
| Randomization | For each Brassica oleracea accession, the sampling process for genome DNA/RNA sequencing was randomly conducted. |
| Blinding | Blinding is not necessary for genome sequencing and assembly, since the investigators know which Brassica oleracea accessions they were handing. |

# Behavioural & social sciences study design

All studies must disclose on these points even when the disclosure is negative.

| | |
|---|---|
| Study description | Briefly describe the study type including whether data are quantitative, qualitative, or mixed-methods (e.g. qualitative cross-sectional, quantitative experimental, mixed-methods case study). |
| Research sample | State the research sample (e.g. Harvard university undergraduates, villagers in rural India) and provide relevant demographic |

| | |
|---|---|
| Research sample | *information (e.g. age, sex) and indicate whether the sample is representative. Provide a rationale for the study sample chosen. For studies involving existing datasets, please describe the dataset and source.* |
| Sampling strategy | *Describe the sampling procedure (e.g. random, snowball, stratified, convenience). Describe the statistical methods that were used to predetermine sample size OR if no sample-size calculation was performed, describe how sample sizes were chosen and provide a rationale for why these sample sizes are sufficient. For qualitative data, please indicate whether data saturation was considered, and what criteria were used to decide that no further sampling was needed.* |
| Data collection | *Provide details about the data collection procedure, including the instruments or devices used to record the data (e.g. pen and paper, computer, eye tracker, video or audio equipment) whether anyone was present besides the participant(s) and the researcher, and whether the researcher was blind to experimental condition and/or the study hypothesis during data collection.* |
| Timing | *Indicate the start and stop dates of data collection. If there is a gap between collection periods, state the dates for each sample cohort.* |
| Data exclusions | *If no data were excluded from the analyses, state so OR if data were excluded, provide the exact number of exclusions and the rationale behind them, indicating whether exclusion criteria were pre-established.* |
| Non-participation | *State how many participants dropped out/declined participation and the reason(s) given OR provide response rate OR state that no participants dropped out/declined participation.* |
| Randomization | *If participants were not allocated into experimental groups, state so OR describe how participants were allocated to groups, and if allocation was not random, describe how covariates were controlled.* |

# Ecological, evolutionary & environmental sciences study design

All studies must disclose on these points even when the disclosure is negative.

| | |
|---|---|
| Study description | *Briefly describe the study. For quantitative data include treatment factors and interactions, design structure (e.g. factorial, nested, hierarchical), nature and number of experimental units and replicates.* |
| Research sample | *Describe the research sample (e.g. a group of tagged Passer domesticus, all Stenocereus thurberi within Organ Pipe Cactus National Monument), and provide a rationale for the sample choice. When relevant, describe the organism taxa, source, sex, age range and any manipulations. State what population the sample is meant to represent when applicable. For studies involving existing datasets, describe the data and its source.* |
| Sampling strategy | *Note the sampling procedure. Describe the statistical methods that were used to predetermine sample size OR if no sample-size calculation was performed, describe how sample sizes were chosen and provide a rationale for why these sample sizes are sufficient.* |
| Data collection | *Describe the data collection procedure, including who recorded the data and how.* |
| Timing and spatial scale | *Indicate the start and stop dates of data collection, noting the frequency and periodicity of sampling and providing a rationale for these choices. If there is a gap between collection periods, state the dates for each sample cohort. Specify the spatial scale from which the data are taken* |
| Data exclusions | *If no data were excluded from the analyses, state so OR if data were excluded, describe the exclusions and the rationale behind them, indicating whether exclusion criteria were pre-established.* |
| Reproducibility | *Describe the measures taken to verify the reproducibility of experimental findings. For each experiment, note whether any attempts to repeat the experiment failed OR state that all attempts to repeat the experiment were successful.* |
| Randomization | *Describe how samples/organisms/participants were allocated into groups. If allocation was not random, describe how covariates were controlled. If this is not relevant to your study, explain why.* |
| Blinding | *Describe the extent of blinding used during data acquisition and analysis. If blinding was not possible, describe why OR explain why blinding was not relevant to your study.* |

Did the study involve field work? ☐ Yes ☐ No

## Field work, collection and transport

| | |
|---|---|
| Field conditions | *Describe the study conditions for field work, providing relevant parameters (e.g. temperature, rainfall).* |
| Location | *State the location of the sampling or experiment, providing relevant parameters (e.g. latitude and longitude, elevation, water depth).* |
| Access & import/export | *Describe the efforts you have made to access habitats and to collect and import/export your samples in a responsible manner and in compliance with local, national and international laws, noting any permits that were obtained (give the name of the issuing authority, the date of issue, and any identifying information).* |

| Disturbance | *Describe any disturbance caused by the study and how it was minimized.* |

# Reporting for specific materials, systems and methods

We require information from authors about some types of materials, experimental systems and methods used in many studies. Here, indicate whether each material, system or method listed is relevant to your study. If you are not sure if a list item applies to your research, read the appropriate section before selecting a response.

## Materials & experimental systems

| n/a | Involved in the study |
|---|---|
| ☒ | ☐ Antibodies |
| ☒ | ☐ Eukaryotic cell lines |
| ☒ | ☐ Palaeontology and archaeology |
| ☒ | ☐ Animals and other organisms |
| ☒ | ☐ Clinical data |
| ☒ | ☐ Dual use research of concern |
| ☐ | ☒ Plants |

## Methods

| n/a | Involved in the study |
|---|---|
| ☒ | ☐ ChIP-seq |
| ☒ | ☐ Flow cytometry |
| ☒ | ☐ MRI-based neuroimaging |

## Antibodies

| Antibodies used | *Describe all antibodies used in the study; as applicable, provide supplier name, catalog number, clone name, and lot number.* |
| Validation | *Describe the validation of each primary antibody for the species and application, noting any validation statements on the manufacturer's website, relevant citations, antibody profiles in online databases, or data provided in the manuscript.* |

## Eukaryotic cell lines

Policy information about cell lines and Sex and Gender in Research

| Cell line source(s) | *State the source of each cell line used and the sex of all primary cell lines and cells derived from human participants or vertebrate models.* |
| Authentication | *Describe the authentication procedures for each cell line used OR declare that none of the cell lines used were authenticated.* |
| Mycoplasma contamination | *Confirm that all cell lines tested negative for mycoplasma contamination OR describe the results of the testing for mycoplasma contamination OR declare that the cell lines were not tested for mycoplasma contamination.* |
| Commonly misidentified lines (See ICLAC register) | *Name any commonly misidentified cell lines used in the study and provide a rationale for their use.* |

## Palaeontology and Archaeology

| Specimen provenance | *Provide provenance information for specimens and describe permits that were obtained for the work (including the name of the issuing authority, the date of issue, and any identifying information). Permits should encompass collection and, where applicable, export.* |
| Specimen deposition | *Indicate where the specimens have been deposited to permit free access by other researchers.* |
| Dating methods | *If new dates are provided, describe how they were obtained (e.g. collection, storage, sample pretreatment and measurement), where they were obtained (i.e. lab name), the calibration program and the protocol for quality assurance OR state that no new dates are provided.* |

☐ Tick this box to confirm that the raw and calibrated dates are available in the paper or in Supplementary Information.

| Ethics oversight | *Identify the organization(s) that approved or provided guidance on the study protocol, OR state that no ethical approval or guidance was required and explain why not.* |

Note that full information on the approval of the study protocol must also be provided in the manuscript.

# Animals and other research organisms

Policy information about [studies involving animals](); [ARRIVE guidelines]() recommended for reporting animal research, and [Sex and Gender in Research]()

| Laboratory animals | *For laboratory animals, report species, strain and age OR state that the study did not involve laboratory animals.* |
| --- | --- |
| Wild animals | *Provide details on animals observed in or captured in the field; report species and age where possible. Describe how animals were caught and transported and what happened to captive animals after the study (if killed, explain why and describe method; if released, say where and when) OR state that the study did not involve wild animals.* |
| Reporting on sex | *Indicate if findings apply to only one sex; describe whether sex was considered in study design, methods used for assigning sex. Provide data disaggregated for sex where this information has been collected in the source data as appropriate; provide overall numbers in this Reporting Summary. Please state if this information has not been collected. Report sex-based analyses where performed, justify reasons for lack of sex-based analysis.* |
| Field-collected samples | *For laboratory work with field-collected samples, describe all relevant parameters such as housing, maintenance, temperature, photoperiod and end-of-experiment protocol OR state that the study did not involve samples collected from the field.* |
| Ethics oversight | *Identify the organization(s) that approved or provided guidance on the study protocol, OR state that no ethical approval or guidance was required and explain why not.* |

Note that full information on the approval of the study protocol must also be provided in the manuscript.

# Clinical data

Policy information about [clinical studies]()
All manuscripts should comply with the ICMJE [guidelines for publication of clinical research]() and a completed [CONSORT checklist]() must be included with all submissions.

| Clinical trial registration | *Provide the trial registration number from ClinicalTrials.gov or an equivalent agency.* |
| --- | --- |
| Study protocol | *Note where the full trial protocol can be accessed OR if not available, explain why.* |
| Data collection | *Describe the settings and locales of data collection, noting the time periods of recruitment and data collection.* |
| Outcomes | *Describe how you pre-defined primary and secondary outcome measures and how you assessed these measures.* |

# Dual use research of concern

Policy information about [dual use research of concern]()

## Hazards

Could the accidental, deliberate or reckless misuse of agents or technologies generated in the work, or the application of information presented in the manuscript, pose a threat to:

| No | Yes | |
| --- | --- | --- |
| ☐ | ☐ | Public health |
| ☐ | ☐ | National security |
| ☐ | ☐ | Crops and/or livestock |
| ☐ | ☐ | Ecosystems |
| ☐ | ☐ | Any other significant area |

## Experiments of concern

Does the work involve any of these experiments of concern:

| No | Yes | |
|---|---|---|
| ☐ | ☐ | Demonstrate how to render a vaccine ineffective |
| ☐ | ☐ | Confer resistance to therapeutically useful antibiotics or antiviral agents |
| ☐ | ☐ | Enhance the virulence of a pathogen or render a nonpathogen virulent |
| ☐ | ☐ | Increase transmissibility of a pathogen |
| ☐ | ☐ | Alter the host range of a pathogen |
| ☐ | ☐ | Enable evasion of diagnostic/detection modalities |
| ☐ | ☐ | Enable the weaponization of a biological agent or toxin |
| ☐ | ☐ | Any other potentially harmful combination of experiments and agents |

## Plants

| | |
|---|---|
| Seed stocks | *Report on the source of all seed stocks or other plant material used. If applicable, state the seed stock centre and catalogue number. If plant specimens were collected from the field, describe the collection location, date and sampling procedures.* |
| Novel plant genotypes | *Describe the methods by which all novel plant genotypes were produced. This includes those generated by transgenic approaches, gene editing, chemical/radiation-based mutagenesis and hybridization. For transgenic lines, describe the transformation method, the number of independent lines analyzed and the generation upon which experiments were performed. For gene-edited lines, describe the editor used, the endogenous sequence targeted for editing, the targeting guide RNA sequence (if applicable) and how the editor was applied.* |
| Authentication | *Describe any authentication procedures for each seed stock used or novel genotype generated. Describe any experiments used to assess the effect of a mutation and, where applicable, how potential secondary effects (e.g. second site T-DNA insertions, mosiacism, off-target gene editing) were examined.* |

## ChIP-seq

### Data deposition

☐ Confirm that both raw and final processed data have been deposited in a public database such as GEO.

☐ Confirm that you have deposited or provided access to graph files (e.g. BED files) for the called peaks.

| | |
|---|---|
| Data access links<br>*May remain private before publication.* | *For "Initial submission" or "Revised version" documents, provide reviewer access links.  For your "Final submission" document, provide a link to the deposited data.* |
| Files in database submission | *Provide a list of all files available in the database submission.* |
| Genome browser session<br>(e.g. UCSC) | *Provide a link to an anonymized genome browser session for "Initial submission" and "Revised version" documents only, to enable peer review.  Write "no longer applicable" for "Final submission" documents.* |

### Methodology

| | |
|---|---|
| Replicates | *Describe the experimental replicates, specifying number, type and replicate agreement.* |
| Sequencing depth | *Describe the sequencing depth for each experiment, providing the total number of reads, uniquely mapped reads, length of reads and whether they were paired- or single-end.* |
| Antibodies | *Describe the antibodies used for the ChIP-seq experiments; as applicable, provide supplier name, catalog number, clone name, and lot number.* |
| Peak calling parameters | *Specify the command line program and parameters used for read mapping and peak calling, including the ChIP, control and index files used.* |
| Data quality | *Describe the methods used to ensure data quality in full detail, including how many peaks are at FDR 5% and above 5-fold enrichment.* |
| Software | *Describe the software used to collect and analyze the ChIP-seq data. For custom code that has been deposited into a community repository, provide accession details.* |

# Flow Cytometry

## Plots

Confirm that:

☐ The axis labels state the marker and fluorochrome used (e.g. CD4-FITC).

☐ The axis scales are clearly visible. Include numbers along axes only for bottom left plot of group (a 'group' is an analysis of identical markers).

☐ All plots are contour plots with outliers or pseudocolor plots.

☐ A numerical value for number of cells or percentage (with statistics) is provided.

## Methodology

Sample preparation | *Describe the sample preparation, detailing the biological source of the cells and any tissue processing steps used.*

Instrument | *Identify the instrument used for data collection, specifying make and model number.*

Software | *Describe the software used to collect and analyze the flow cytometry data. For custom code that has been deposited into a community repository, provide accession details.*

Cell population abundance | *Describe the abundance of the relevant cell populations within post-sort fractions, providing details on the purity of the samples and how it was determined.*

Gating strategy | *Describe the gating strategy used for all relevant experiments, specifying the preliminary FSC/SSC gates of the starting cell population, indicating where boundaries between "positive" and "negative" staining cell populations are defined.*

☐ Tick this box to confirm that a figure exemplifying the gating strategy is provided in the Supplementary Information.

# Magnetic resonance imaging

## Experimental design

Design type | *Indicate task or resting state; event-related or block design.*

Design specifications | *Specify the number of blocks, trials or experimental units per session and/or subject, and specify the length of each trial or block (if trials are blocked) and interval between trials.*

Behavioral performance measures | *State number and/or type of variables recorded (e.g. correct button press, response time) and what statistics were used to establish that the subjects were performing the task as expected (e.g. mean, range, and/or standard deviation across subjects).*

## Acquisition

Imaging type(s) | *Specify: functional, structural, diffusion, perfusion.*

Field strength | *Specify in Tesla*

Sequence & imaging parameters | *Specify the pulse sequence type (gradient echo, spin echo, etc.), imaging type (EPI, spiral, etc.), field of view, matrix size, slice thickness, orientation and TE/TR/flip angle.*

Area of acquisition | *State whether a whole brain scan was used OR define the area of acquisition, describing how the region was determined.*

Diffusion MRI ☐ Used ☐ Not used

## Preprocessing

Preprocessing software | *Provide detail on software version and revision number and on specific parameters (model/functions, brain extraction, segmentation, smoothing kernel size, etc.).*

Normalization | *If data were normalized/standardized, describe the approach(es): specify linear or non-linear and define image types used for transformation OR indicate that data were not normalized and explain rationale for lack of normalization.*

Normalization template | *Describe the template used for normalization/transformation, specifying subject space or group standardized space (e.g. original Talairach, MNI305, ICBM152) OR indicate that the data were not normalized.*

Noise and artifact removal | *Describe your procedure(s) for artifact and structured noise removal, specifying motion parameters, tissue signals and physiological signals (heart rate, respiration).*

| Volume censoring | *Define your software and/or method and criteria for volume censoring, and state the extent of such censoring.* |
|---|---|

## Statistical modeling & inference

| Model type and settings | *Specify type (mass univariate, multivariate, RSA, predictive, etc.) and describe essential details of the model at the first and second levels (e.g. fixed, random or mixed effects; drift or auto-correlation).* |
|---|---|
| Effect(s) tested | *Define precise effect in terms of the task or stimulus conditions instead of psychological concepts and indicate whether ANOVA or factorial designs were used.* |

Specify type of analysis: ☐ Whole brain ☐ ROI-based ☐ Both

| Statistic type for inference<br><br>(See Eklund et al. 2016) | *Specify voxel-wise or cluster-wise and report all relevant parameters for cluster-wise methods.* |
|---|---|
| Correction | *Describe the type of correction and how it is obtained for multiple comparisons (e.g. FWE, FDR, permutation or Monte Carlo).* |

## Models & analysis

| n/a | Involved in the study |
|---|---|
| ☐ | ☐ Functional and/or effective connectivity |
| ☐ | ☐ Graph analysis |
| ☐ | ☐ Multivariate modeling or predictive analysis |

| Functional and/or effective connectivity | *Report the measures of dependence used and the model details (e.g. Pearson correlation, partial correlation, mutual information).* |
|---|---|
| Graph analysis | *Report the dependent variable and connectivity measure, specifying weighted graph or binarized graph, subject- or group-level, and the global and/or node summaries used (e.g. clustering coefficient, efficiency, etc.).* |
| Multivariate modeling and predictive analysis | *Specify independent variables, features extraction and dimension reduction, model, training and evaluation metrics.* |

