## [Peer Review File · Nature Genetics]

Peer Review Information

Manuscript Title: Large-scale Gene Expression Alterations Introduced by Structural Variation Drive Morphotype Diversification in Brassica oleracea

Corresponding author name(s): Feng Cheng, Yangyong Zhang, Guusje Bonnema, Honghao Lv, Kang Zhang

Editorial Notes:

Transferred manuscripts This document only contains reviewer comments, rebuttal and decision letters for versions considered at Nature Genetics.

Reviewer Comments & Decisions:

Decision Letter, initial version:

18th Jul 2023

Dear Dr Cheng,

Your Article, "Large-scale Gene Expression Variation Introduced by Structural Variation Drives Morphotype Diversification in Brassica oleracea" has now been seen by 2 referees. You will see from their comments below that while they find your work of interest, some important points are raised. We are interested in the possibility of publishing your study in Nature Genetics, but would like to consider your response to these concerns in the form of a revised manuscript before we make a final decision on publication.

To guide the scope of the revisions, the editors discuss the referee reports in detail within the team with a view to identifying key priorities that should be addressed in revision. In this case, we think both referees have provided constructive reviews aimed at strengthening the analyses and improving the presentation, and we particularly ask that you address their technical comments as thoroughly as possible with appropriate revisions. We hope that you will find the prioritized set of referee points to be useful when revising your study. Please do not hesitate to get in touch if you would like to discuss these issues further.

We therefore invite you to revise your manuscript taking into account all reviewer and editor comments. Please highlight all changes in the manuscript text file. At this stage we will need you to upload a copy of the manuscript in MS Word .docx or similar editable format.

*2) If you have not done so already please begin to revise your manuscript so that it conforms to our Article format instructions, available [here](http://www.nature.com/ng/authors/article_types/index.html). Refer also to any guidelines provided in this letter.

[redacted]

We hope to receive your revised manuscript within 3 to 6 months. If you cannot send it within this time, please let us know.

Sincerely,
Wei

Wei Li, PhD
Senior Editor
Nature Genetics
New York, NY 10004, USA
www.nature.com/ng

Reviewers' Comments:

Reviewer #1:

Remarks to the Author:

The manuscript 'Large-scale Gene Expression Variation Introduced by Structural Variation 1 Drives Morphotype Diversification in *Brassica oleracea*' by Li et al. described the sequencing of several *B. oleracea* genomes, the construction of a graph pangenome and the analysis of structural variation, particularly in relation to gene expression and morphological traits. *B. oleracea* is an excellent model for such analysis, demonstrating diverse morphology associated with its use and as an important food crop.

The authors identified structural variation associated with several important traits, some of which was already known, providing confidence in their analysis, while providing a broader perspective of how structural variation impacts traits. The manuscript is well written and is a significant advance on other recent pangenome studies. The study was possible due to recent advances in genome sequencing technology and the authors have applied the latest and most robust methods for their analysis. The links to metabolism associated with different morphotypes is of particular novelty and interest.

I have only a few minor suggestions on how the manuscript could be improved further.

Line 264: 93% and 97% of long and short SVs could be validated. It isn't clear whether the other 7% and 3% were shown to be false SV calls or if they just could not be validated. This should be clarified.

Line 270: the number of SVs increases with the number of genomes. Would it be possible to use this to predict the total number of SVs in the species? This would support the statement on line 534 that the study provides the 'full landscape'.

Line 274: reports 27 'core' SVs. This seems counter intuitive as if all individuals had the SV (defined as core) there would be no SV?

Line 276: As SVs are variants between one or more individual, it isn't clear how SVs per genome was calculated. If they are calculated by comparison with a single reference, this would introduce bias in

calculating this number. This point should be clarified.

Line 300: is the difference between SV genes suppressing or promoting expression significant?

Line 414: only 4 of the lines had the insertion in cabbage accessions, was there anything particular in the morphology of these four? How can these four be explained?

Line 488, regarding discussion on the presence of the SV genotype decreasing expression, it may be more clear if this was described as the presence of an insertion or absence of deletion rather than a more generic SV which could be an insertion or deletion of DNA.

Line 505: It is well understood that sequence variation is reduced in genic regions due to selective pressure/the need for these sequences to encode a functional protein so this paragraph could be simplified to acknowledge this fact.

Figure 3E: the % SV frequency increases from left to right. Does this reflect moving away from the coding sequence in the up and downstream regions? Why do we see this in exons and intergenic regions?

Figure 3F. Why does the red line (up and downstream regions) cover the gene body region?

The reference to NCBI does not seem to link to any data.

Reviewer #2:

Remarks to the Author:

Li et al. assembled 22 genomes in *B. oleracea* and revealed structural variations across these genomes and five previously reported genomes. They first explored the distribution of transposable elements, Copia and Gypsy, on the genomes, to demonstrate that transposable elements played a role in the genomic variation. Next, they separated genes on the whole genome into different groups and characterized each group by number of TEs, expression levels, number of homoeologs. Specifically, they conducted functional analysis on private genes, relating these genes into multiple metabolite pathways. Next, they extracted structural variations (SVs) from the 27 genomes and associated PAVs (one type of SV) with expression levels of nearby genes. They concluded that SVs increased nearby gene expressions likely through introducing extra transcription factor binding sites, and decreased gene expressions likely through the rising methylation levels. These conclusions were validated by cases of several genes. The core idea of this manuscript was that SVs altered gene expression, resulting in different phenotypes, through selection, shaping diverse morphotypes.

This manuscript is of interest and importance and will draw much attention as it consolidated the role of SV in crop domestication and phenotype diversification. Still, I have some questions that need clarification.

1) Authors did not use any SNP data, and did not even discuss SNP. Is the contribution of SVs to morphotype diversity larger than those of SNP? Could the genes detected with SVs also be detected by SNPs?

2) The association between PAV and expression level of nearby gene was conducted by using 27 genomes. For each PAV-gene pair, the sample size is 27. Why not to use data from 223 accessions

since authors collected the mRNA-seq data from these accessions? The size of 223 was much larger than of 27, the confidence of association should raise. Were results of SV-gene association from the 223-dataset consistent with those from 27-dataset?

3) Still with the 223-dataset, why not to use this dataset to conduct EWAS, in which gene expression is regarded as phenotype and SV as genotype? This analysis might result in SV-gene association in trans, i.e., SV regulated expression of a distant gene.

4) In figure 4A, authors showed that number of SV genes decreased when SV was far away from gene. Was there a result showing that gene expression level was influenced less when SV was far away from gene? Was the difference of expression between gene with an SV and syntenic gene without an SV large when SV was close to gene and small when SV was distant?

5) In line 259, authors extracted 13,090 copy number variations, why not to uncover the association between the copy number and gene expression level? It could be more interesting to use copy number than PAV because copy number is considered as an ordinal factor while PAV is a discrete and biallelic factor.

6) In line 513, there were two "indeed". In acknowledgement, what is "CC"?

Author Rebuttal to Initial comments

Point by point responses to reviewers' comments

Reviewer #1:

The manuscript 'Large-scale Gene Expression Variation Introduced by Structural Variation Drives Morphotype Diversification in *Brassica oleracea*' by Li et al. described the sequencing of several *B. oleracea* genomes, the construction of a graph pangenome and the analysis of structural variation, particularly in relation to gene expression and morphological traits. *B. oleracea* is an excellent model for such analysis, demonstrating diverse morphology associated with its use and as an important food crop.

The authors identified structural variation associated with several important traits, some of which was already known, providing confidence in their analysis, while providing a broader perspective of how structural variation impacts traits. The manuscript is well written and is a significant advance on other recent pangenome studies. The study was possible due to recent advances in genome sequencing technology and the authors have applied the latest and most robust methods for their analysis. The links to metabolism associated with different morphotypes is of particular novelty and interest.

I have only a few minor suggestions on how the manuscript could be improved further.

Authors: First of all, thank you very much for your positive comments and constructive advice for us to improve our manuscript. We carefully revised our manuscript and added more methods in detail according to your suggestions. We hope that we have addressed all your

questions in the updated ms.

Q1: Line 264: 93% and 97% of long and short SVs could be validated. It isn't clear whether the other 7% and 3% were shown to be false SV calls or if they just could not be validated. This should be clarified.

A1: We are sorry for the confusion. For the long SVs, the remaining 7% could not be validated by Hi-C data; only weak interaction signals on both the SVs and their flanking sequences were detected. These SVs were located around the centromere regions. While for the short SVs, the remaining 3% were shown to be false calls, based on the long-reads alignments and further checks with the visualization in IGV. We revised this sentence to clarify the results.

Revisions (in red) in Results lines 265-267: Approximately 93% of the selected large SVs were validated by Hi-C paired-end reads, the remaining 7% could not be validated. However for the selected short SVs, 97% were validated by long-reads, while 3% were found to be false calls.

Q2: Line 277: the number of SVs increases with the number of genomes. Would it be possible to use this to predict the total number of SVs in the species? This would support the statement on line 534 that the study provides the 'full landscape'.

A2: Thank you for your suggestion. We have performed the prediction of the total number of SVs in *B. oleracea* population using previously reported method, and added the content in the updated manuscript.

Revisions in Results line 274: Modelling this increase predicts a total SV size of 58,410 ± 1,452.

Modelling method citation: "Hurgobin, B. et al. Homoeologous exchange is a major cause of gene presence/absence variation in the amphidiploid Brassica napus. Plant Biotechnol J 16, 1265-1274 (2018)".

Q3: Line 274: reports 27 'core' SVs. This seems counter intuitive as if all individuals had the SV (defined as core) there would be no SV?

A3: Sorry for the confusion. The number 27 was the number of SV loci, but did not refer to particular SVs that are present in 27 genomes. These 27 'core' SVs were found in all the other 26 genomes when compared to the reference T10.

Q4: Line 276: As SVs are variants between one or more individual, it isn't clear how SVs per

genome was calculated. If they are calculated by comparison with a single reference, this would introduce bias in calculating this number. This point should be clarified.

A4: Yes, we agree with the reviewer that SVs calculated by comparison with a single reference would introduce bias, especially in calculating/comparing SV numbers between different genomes. Therefore, we removed the sentence of line 276 in the updated manuscript. Before constructing the graph-based genome to determine SVs in the 704 *B. oleracea* accessions, we took the wild *B. oleracea* T10 genome as the reference and aligned the other 26 *B. oleracea* genomes to T10 to identify SVs in these genomes. The number and distribution of these identified SVs illustrate the general features of SVs in the pan-genome of *B. oleracea*. Similar analysis were performed in the pan-genome studies of rice and soybean, etc.

The rice pangenome: "Qin, P. et al. Pan-genome analysis of 33 genetically diverse rice accessions reveals hidden genomic variations. Cell 184, 3542-3558 e16 (2021)". The soybean pangenome: "Liu, Y. et al. Pan-Genome of Wild and Cultivated Soybeans. Cell 182, 162-176 e13 (2020)".

Q5: Line 300: is the difference between SV genes suppressing or promoting expression significant?

A5: Yes, the difference between SV genes suppressing or promoting expression is significant. We added the statistical test in the updated manuscript.

Revisions (in red) in Results lines 301-304: Of these 6,526 SV genes, SV presence was associated with suppressed expression in 3,536 (54%) SV genes and promoted expression in 2,990 (46%) SV genes (Figure 4B), with significantly more SV genes showing suppressed than promoted expression (P value = 1.48×10^{-11} , binomial test).

Q6: Line 414: only 4 of the lines had the insertion in cabbage accessions, was there anything particular in the morphology of these four? How can these four be explained?

A6: We checked these four cabbage accessions (Sample ID: S1, GL89, TKI816, and cabbage_029) with the presence (insertion) genotype; they were heading accessions showing somewhat loose leafy head morphology, surveyed in a field experiment in autumn 2022 in Beijing comparing to other cabbages, but this was a characteristic of more cabbage accessions. As leafy head formation is a complex trait with many genes involved, the lack of this specific variation may be compensated by other variations.

Q7: Line 488, regarding discussion on the presence of the SV genotype decreasing expression,

it may be more clear if this was described as the presence of an insertion or absence of deletion rather than a more generic SV which could be an insertion or deletion of DNA.

A7: Yes, we agree with your idea. Actually, in order to avoid this confusion in referring to the genotype of 'Insertion' or 'Deletion', we previously defined the two terms 'presence' and 'absence' in lines 292-294: "*To be independent of the reference genome used for SV calling, we defined the genotype with insertion (more sequence) as 'presence' and the genotype without (thus less sequence) as 'absence'.*". For line 488, we also revised "the presence SV genotype" into "the presence genotype" to make it clearer.

Q8: Line 505: It is well understood that sequence variation is reduced in genic regions due to selective pressure/the need for these sequences to encode a functional protein so this paragraph could be simplified to acknowledge this fact.

A8: Thanks for the advice, we revised this paragraph to simplify it as suggested.

Revisions (in red) in Discussion lines 540-544: SV frequency was much higher in promotor and downstream regions of genes than in the exons and introns. Compared to SVs that disrupt coding sequences, which are under strong selection pressure, SVs located in promoter/downstream regions do not affect gene function, but exert moderate effects on gene activity by changing their expression dosage, and are therefore likely to be under less stringent selection pressure and can accumulate in genomes.

Q9: Figure 3E: the % SV frequency increases from left to right. Does this reflect moving away from the coding sequence in the up and downstream regions? Why do we see this in exons and intergenic regions?

A9: I'm sorry for the confusion. In the maintext **Figure 3E**, we calculated the ratio of SVs that located in five different genomic regions as gene upstream (within -3 kb), exon, intron, gene downstream (within +3 kb), and intergenic (the left regions) for each of the 27 genomes. Then, for each of the five regions, the values (SV ratio) of the 27 genomes were sorted and plotted from small to large. We also revised the figure legend to make it clearer.

*Revisions (in red) in maintext **Figure 3 (E)** legend lines 791-794: The frequency distribution of SVs in five different genomic regions around the genes: upstream (within -3 kb), exon, intron, downstream (within +3 kb), and intergenic for the 27 B. oleracea genomes. The SV ratios in the five regions were calculated for each of the 27 genomes, these values were then sorted and plotted from small to large for each of the five regions.*

Q10: Figure 3F. Why does the red line (up and downstream regions) cover the gene body region?

A10: The red line in the gene-body denoted the density of SV sequences located at genes (exon and intron). We deleted the red line covering the gene-body region to avoid confusion in the updated manuscript.

Q11: The reference to NCBI does not seem to link to any data.

A11: We have uploaded all the datasets into the public repertoire, they will be freely available upon acceptance of the manuscript.

Reviewer #2:

Li et al. assembled 22 genomes in *B. oleracea* and revealed structural variations across these genomes and five previously reported genomes. They first explored the distribution of transposable elements, Copia and Gypsy, on the genomes, to demonstrate that transposable elements played a role in the genomic variation. Next, they separated genes on the whole genome into different groups and characterized each group by number of TEs, expression levels, number of homoeologs. Specifically, they conducted functional analysis on private genes, relating these genes into multiple metabolite pathways. Next, they extracted structural variations (SVs) from the 27 genomes and associated PAVs (one type of SV) with expression levels of nearby genes. They concluded that SVs increased nearby gene expressions likely through introducing extra transcription factor binding sites, and decreased gene expressions likely through the rising methylation levels. These conclusions were validated by cases of several genes. The core idea of this manuscript was that SVs altered gene expression, resulting in different phenotypes, through selection, shaping diverse morphotypes.

This manuscript is of interest and importance and will draw much attention as it consolidated the role of SV in crop domestication and phenotype diversification. Still, I have some questions that need clarification.

Authors: Thank you very much for your positive comments and constructive advice. We carefully added new analyses and revised the ms. based on your suggestions. We hope that all your questions have been addressed in this updated ms.

Q1: Authors did not use any SNP data, and did not even discuss SNP. Is the contribution of SVs to morphotype diversity larger than those of SNP? Could the genes detected with SVs also be detected by SNPs?

A1: We previously analyzed the SNP data, but deleted these contents when we decided to focus on the SV-related results. In the updated manuscript, we added back the SNP-related contents

(Lines 357-363 and Supplementary Figure 10). The story of our manuscript introduces SVs affecting gene expression patterns as one major force in the domestication and diversification of *B. oleracea*. Of course, we do not exclude the existence of other mechanisms, such as SVs affecting protein-coding sequences of genes, as well as SNPs and InDels. We have added this discussion in the revised manuscript.

SNPs are important contributors to genetic variation. However, in this study, we find that SVs can not be completely represented by SNPs, and SNPs did not play an equal role in affecting gene expression compared to that of SVs. We performed linkage disequilibrium (LD) analysis between SV and SNPs, and found that more than half of these SVs (54.78%) showed low LD ($r^2 < 0.5$) with SNPs, indicating that the contribution of a considerable part of SVs to the morphotype diversity, as well as the expression variation of causal genes, could not be detected by SNPs.

In addition, of the five genes provided as examples in our manuscript, one gene, *BoMYB*, cannot be detected by SNPs, whereas the other four genes (*BoKANI*, *BoACS4*, *BoPNY*, and *BoCKX3*) can be detected by SNPs. However, the four SNPs with the strongest signals are located in the flanking regions of the four corresponding genes, and thus do not affect their coding sequences. Furthermore, these SNPs show high LD ($r^2 = 0.97, 1.00, 1.00, \text{ and } 0.98$, respectively) with corresponding SVs detected previously, indicating that these SNPs should be the hitchhikers of SV signals affecting gene expression.

Revisions in Discussion lines 563-567: Adding to this, even though the current study found that SVs were a major force in B. oleracea domestication/diversification by affecting gene expression, it does not exclude other mechanisms, such as SVs affecting protein-coding sequences of genes, as well as SNPs and InDels, in the domestication of B. oleracea crops.

Q2: The association between PAV and expression level of nearby genes was conducted by using 27 genomes. For each PAV-gene pair, the sample size is 27. Why not to use data from 223 accessions since authors collected the mRNA-seq data from these accessions? The size of 223 was much larger than of 27, the confidence of association should raise. Were results of SV-gene association from the 223-dataset consistent with those from 27-dataset?

A2: We thank the reviewer for the constructive comments. Yes, we initially identified SVs among these 27 genomes by direct alignment and comparison of their genome sequences. Using these SVs, we analyzed the relationships between SVs and the expression of their nearby genes in the 27 genomes. We agree with the reviewer that more samples with mRNA-seq data will increase confidence. Therefore, in the updated manuscript, after calling the SV

genotypes in the *B. oleracea* population using the graph-based genome and resequencing data of 704 accessions, we performed association analysis between SV and gene expression using the 223 accessions for which mRNA-seq data were available.

In total, there were 7,685 SV genes in the *B. oleracea* population, of which 4,366 SV genes were expressed based on the mRNA-seq data of 223 accessions. For each SV locus, we checked that at least 60 samples were successfully genotyped in the 223 accessions, with at least ten samples present in each of the two genotype groups. After filtering, we obtained 3,216 SV genes for further analysis. These 3,216 SV genes were divided into five groups as in the analysis of the 27 genomes. The percentage of SVs associated with the expression of the corresponding genes ranged from 68% for SVs located in the gene-body to 59% for SVs located 5-10 kb away from the genes. In total, 61% of these SVs were significantly (P value < 0.05) associated with the expression of the corresponding SV genes, which is slightly less than the analysis using 27 genomes (69%). Of these significantly associated SV genes, the presence genotypes of SVs were significantly (P value < 0.05) associated with the suppressed expression of 1,071 (55%) genes, and with the promoted expression of 888 (45%) genes, consistent with the 27-genome analysis (54% suppression, 46% promotion). We have included these results in the updated manuscript.

*Revisions in Results lines 364-378: Based on SV data from 704 accessions and mRNA-seq data from 223 representative accessions (Supplementary Table 1), we further investigated the relationship between SV and gene expression. Of the 7,685 SV genes in the *B. oleracea* population, 4,366 SV genes were expressed and 3,216 were filtered for further analysis (Methods). These 3,216 SV genes were divided into five groups as aforementioned. The results showed that the percentage of SVs significantly (P value < 0.05) associated with the expression of the corresponding genes ranged from 68% for SVs located in the gene-body to 59% for SVs located 5-10 kb away from the genes. In total, 61% of these SVs were significantly associated with the expression of the corresponding genes, which is slightly less than 69% comparing SV-gene expression between absence/presence groups of the 27 genomes. Of these significantly associated SV genes, the presence genotypes were significantly associated with suppressed expression of 1,071 (55%) genes and promoted expression of 888 (45%) genes, consistent with the 27-genome analysis (54% suppression, 46% promotion).*

Q3: Still with the 223-dataset, why not to use this dataset to conduct EWAS, in which gene expression is regarded as phenotype and SV as genotype? This analysis might result in SV-gene association in trans, i.e., SV regulated expression of a distant gene. **A3:** Thank you for

your suggestion. In our manuscript, we focused on the positional effect of SVs on the expression of their neighbouring genes by identifying SV-gene pairs and analyzing the relationships between SV genotype and gene expression in these SV gene pairs. Based on your suggestion and to make our analysis more complete, in the revised manuscript we performed and added the contents of the eGWAS analysis using all SVs and the expression of all genes.

There were 17,696 expressed genes (with a TPM ≥ 10 in at least 10% of all samples) that were used as traits for the eGWAS analysis. Of these, the expression of 8,180 (46%) genes showed a significant association (P value $< 1.00 \times 10^{-10}$) with at least one SV. In total, we identified 50,076 SVs that were significantly associated with the expression of these 8,180 genes. As shown in **Supplementary Table 16**, of all these signals, 23% (11,536) were intra-chromosome SVs and 77% (38,540) were inter-chromosome SVs. Of these 11,536 intra-chromosome SV signals, 1,335 were *cis*-regulatory SVs (20 kb within upstream or downstream of a gene)—51% were promoting regulators and 49% were suppressing regulators. All of the remaining signals (48,741) were *trans*-regulatory SVs, with an average of six *trans*-regulatory SVs per gene, and with 53% and 47% showing promoting and suppressing regulation, respectively. We have included these results in the updated manuscript.

Revisions in Results lines 379-392: In addition to the SV gene analysis, we also performed the SV-based eGWAS analysis using the mRNA-seq data of the 223 representative accessions. In this eGWAS analysis, 17,696 expressed genes and

*40,028 SVs were used as traits and markers, respectively (Methods). We obtained 8,180 genes whose expression was significantly associated (P value $< 1.00 \times 10^{-10}$) with at least one SV. In total, 86,435 significant SV signals were identified. As shown in **Supplementary Table 16**, 23% (11,536) and 77% (38,540) of these signals were intra- and inter-chromosome SVs, respectively. Of the 11,536 intra-chromosome signals, 1,335 were *cis*-regulatory SVs (20 kb within upstream or downstream of a gene)—with 51% and 49% showing promoting and suppressing regulation, respectively. All of the remaining signals (48,741) were *trans*-regulatory SVs, with an average of six *trans*-regulatory SVs per gene, and with 53% and 47% showing promoting and suppressing regulation, respectively. These results further indicate the importance and complex regulatory role of SVs in gene expression variation.*

Q4: In figure 4A, authors showed that number of SV genes decreased when SV was far away from gene. Was there a result showing that gene expression level was influenced less when SV was far away from gene? Was the difference of expression between gene with an SV and syntenic gene without an SV large when SV was close to gene and small when SV was distant?

A4: Yes, the gene expression level was influenced less when SV was far away from gene. SVs did show a position effect on gene expression, in that the level of gene expression was less affected when the SV was further away from the gene. This trend can be seen in the maintext **Figure 4D**. Here, we performed boxplot based on the data in **Figure 4D** to make this result more observable (below **Figure R1**). The **Figure R1** showed the general trend that when the SV is located far away from gene, the fold-change of gene expression become lower. In addition, as shown in below **Figure R2**, most genes are less than 3 kb from their neighbouring genes in *B. oleracea* genome, so the number of SV genes largely decreased when the SV was far away from gene.

Figure R1. The swarmplot showing the range of fold-changes of SV-gene expression between presence and absence genotype groups along with the distance changing between SV and SV-gene.

Figure R2. The frequency distribution of distance between neighbouring gene pairs in genome of *B. oleracea* (Take T10 genome as example).

Q5: In line 259, authors extracted 13,090 copy number variations, why not to uncover the association between the copy number and gene expression level? It could be more interesting to use copy number than PAV because copy number is considered as an ordinal factor while PAV is a discrete and biallelic factor.

A5: Thank you for your comments. We performed the association analysis between CNVs and the gene expression as suggested.

As referred in our manuscript, these 13,090 CNVs are the sum of the number of CNVs identified between the T10 reference and each of the other 26 genomes. After removing redundancy, we obtained 2,986 CNV loci among the 27 genomes. Using the same rule as in the PAV analysis, we found that there were 490 out of these 2,968 CNVs identified as CNV gene. Of these 490 CNV genes, 352 were expressed. We further found that 106 (30.11%) of the 352 CNVs were significantly associated with the expression of their corresponding CNV genes, with 58 and 48 CNVs showing positive and negative association with gene expression, respectively.

As you can read from above, these CNVs were limited in number, especially those in close

vicinity of genes. As a result, the number/ratio of CNVs that showed association with gene expression is also much lower compared to that of PAVs. For that reason no solid conclusion can be drawn on the CNV data. Therefore, we did not add these CNV gene expression data to the manuscript.

Q6: In line 513, there were two “indeed”. In acknowledgement, what is “CC”? **A6:** Thanks, we removed one “indeed”, and revised CC to C.C., which is short for Chengcheng Cai.

Decision Letter, first revision:

16th Oct 2023

Dear Dr. Cheng,

Thank you for submitting your revised manuscript "Large-scale Gene Expression Variation Introduced by Structural Variation Drives Morphotype Diversification in Brassica oleracea" (NG-A62801R). It has now been seen by the original referees and their comments are below. The reviewers find that the paper has improved in revision, and therefore we'll be happy in principle to publish it in Nature Genetics, pending minor revisions to satisfy the referees' final requests and to comply with our editorial and formatting guidelines.

Sincerely,
Wei

Wei Li, PhD
Senior Editor
Nature Genetics
New York, NY 10004, USA
www.nature.com/ng

Reviewer #1 (Remarks to the Author):

Many thanks for the opportunity to review a revised version of this manuscript. The authors have

addressed each of my comments and the manuscript is improved. I only have only one remaining suggestion. The description of 'core' SV on line 277 could be confusing as the definition is different from 'core' genes. Core genes are present in all lines while 'core' SVs are present or absent only in the reference compared to the other lines. I suggest rewording this sentence without the use of 'core', 'softcore' or 'private' SVs to avoid this confusion. Perhaps define as 'present only in one line', 'present in <10% lines', 'present in >90% lines' and absent in one line', or something similar, to be more precise.

Reviewer #2 (Remarks to the Author):

In the revised manuscript, Li et al. have addressed all the proposed questions. Overall, I have no further inquiries regarding this manuscript, except for one minor issue: the tense should be kept consistence. For example, the authors used present tense in all the subtitles in the Results section, except for one instance where they used past tense in "SV mediated expression dosage alteration associated with morphotypes".

Author Rebuttal, first revision:

Point-by-point responses to reviewers' comments

Reviewer #1:

Remarks to the Author:

Many thanks for the opportunity to review a revised version of this manuscript. The authors have addressed each of my comments and the manuscript is improved. I only have only one remaining suggestion. The description of 'core' SV on line 277 could be confusing as the definition is different from 'core' genes. Core genes are present in all lines while 'core' SVs are present or absent only in the reference compared to the other lines. I suggest rewording this sentence without the use of 'core', 'softcore' or 'private' SVs to avoid this confusion. Perhaps define as 'present only in one line', 'present in <10% lines', 'present in >90% lines' and absent in one line', or something similar, to be more precise.

Response: We thank the reviewer for the supportive comments and suggestion. We have reworded this sentence in the revised manuscript (Pages 8-9, Lines 210-212).

Revisions (in red) made regarding this comment: The number of shared SVs sharply declined for the first three genomes and slowly decreased thereafter. We identified 27 SVs present in all 26

*query genomes, 168 SVs present in 24-25 query genomes, 26,641 SVs present in 2-23 query genomes, and 18,226 SVs present in only one query genome, opposite to the trend of gene family counts (Fig. 3c). The number of private SVs in wild *B. oleracea* is significantly higher than in broccoli/cauliflower and cabbage, indicating extensive loss of genetic diversity during domestication of *B. oleracea* (Figs. 3c and 3d).*

Reviewer #2:

Remarks to the Author:

In the revised manuscript, Li et al. have addressed all the proposed questions. Overall, I have no further inquiries regarding this manuscript, except for one minor issue: the tense should be kept consistent. For example, the authors used present tense in all the subtitles in the Results section, except for one instance where they used past tense in “SV mediated expression dosage alteration associated with morphotypes”.

Response: We thank the reviewer for the supportive comments. We made the tense consistent in the revised version and the subtitle has been changed to the present tense (Page 11, Line 289).

Revisions (in red) made regarding this comment: Expression Alterations by SVs Associate with Morphotypes.

Final Decision Letter:

In reply please quote: NG-A62801R1 Cheng

3rd Jan 2024

Dear Dr. Cheng,

I am delighted to say that your manuscript "Large-scale gene expression alterations introduced by structural variation drive morphotype diversification in *Brassica oleracea*" has been accepted for publication in an upcoming issue of Nature Genetics.

Over the next few weeks, your paper will be copyedited to ensure that it conforms to Nature Genetics style. Once your paper is typeset, you will receive an email with a link to choose the appropriate publishing options for your paper and our Author Services team will be in touch regarding any

additional information that may be required.

Your paper will be published online after we receive your corrections and will appear in print in the next available issue. You can find out your date of online publication by contacting the Nature Press Office (press@nature.com) after sending your e-proof corrections.

Please note that *Nature Genetics* is a Transformative Journal (TJ). Authors may publish their research with us through the traditional subscription access route or make their paper immediately open access through payment of an article-processing charge (APC). Authors will not be required to make a final decision about access to their article until it has been accepted. [Find out more about Transformative Journals](https://www.springernature.com/gp/open-research/transformative-journals)

Authors may need to take specific actions to achieve [compliance](https://www.springernature.com/gp/open-research/funding/policy-compliance-faqs) with funder and institutional open access mandates. If your research is supported by a funder that requires immediate open access (e.g. according to [Plan S principles](https://www.springernature.com/gp/open-research/plan-s-compliance))

then you should select the gold OA route, and we will direct you to the compliant route where possible. For authors selecting the subscription publication route, the journal's standard licensing terms will need to be accepted, including <https://www.nature.com/nature-portfolio/editorial-policies/self-archiving-and-license-to-publish>. Those licensing terms will supersede any other terms that the author or any third party may assert apply to any version of the manuscript.

If you have not already done so, we invite you to upload the step-by-step protocols used in this manuscript to the Protocols Exchange, part of our on-line web resource, natureprotocols.com. If you complete the upload by the time you receive your manuscript proofs, we can insert links in your article that lead directly to the protocol details. Your protocol will be made freely available upon publication of your paper. By participating in natureprotocols.com, you are enabling researchers to more readily reproduce or adapt the methodology you use. [Natureprotocols.com](http://natureprotocols.com) is fully searchable, providing your protocols and paper with increased utility and visibility. Please submit your protocol to <https://protocolexchange.researchsquare.com/>. After entering your [nature.com](http://www.nature.com) username and password you will need to enter your manuscript number (NG-A62801R1). Further information can be found at <https://www.nature.com/nature-portfolio/editorial-policies/reporting-standards#protocols>

Sincerely,
Wei

Wei Li, PhD
Senior Editor
Nature Genetics

New York, NY 10004, USA
www.nature.com/ng